# SCALABLE RANKED PREFERENCE OPTIMIZATION FOR TEXT-TO-IMAGE GENERATION

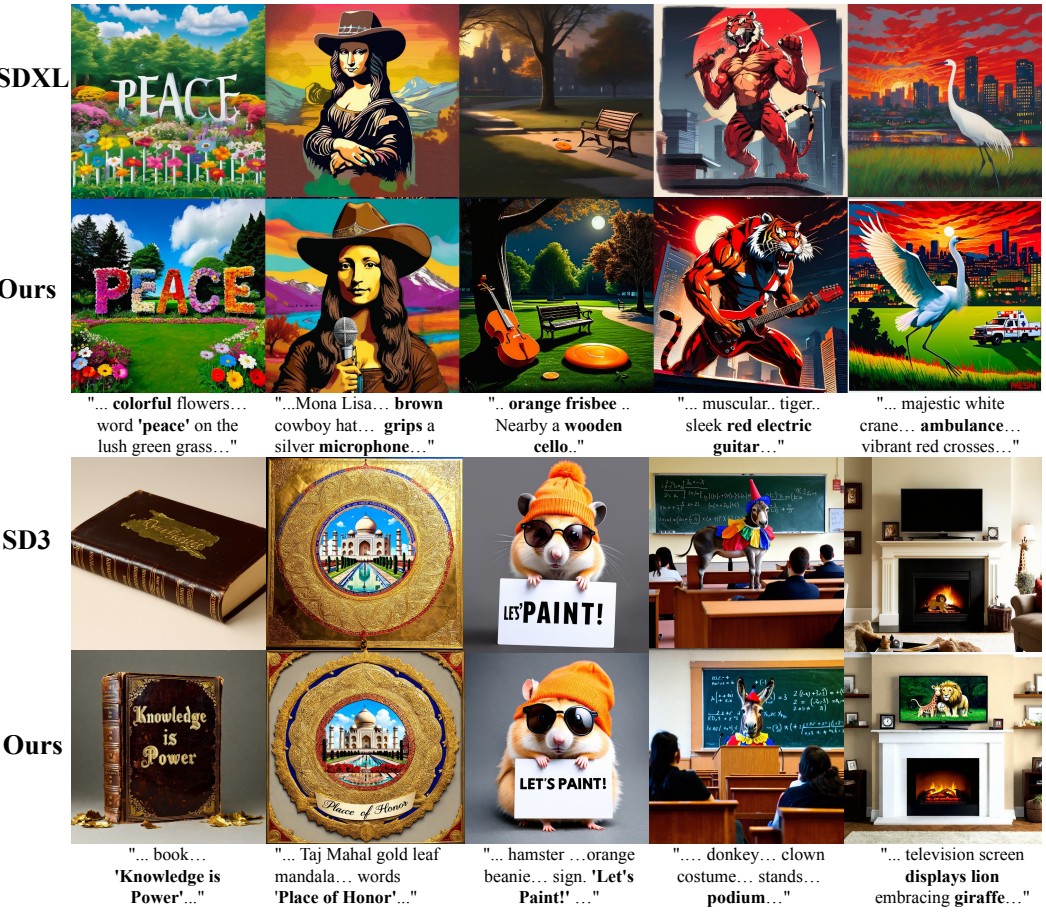

Figure 1: Our approach, trained on a synthetic preference dataset with a ranking objective in the preference optimization, improves *prompt following* and *visual quality* for SDXL (Podell et al., 2023) and SD3-Medium (Esser et al., 2024), without requiring any manual annotations.

## ABSTRACT

Direct Preference Optimization (DPO) has emerged as a powerful approach to align text-to-image (T2I) models with human feedback. Unfortunately, successful application of DPO to T2I models requires a huge amount of resources to collect and label large-scale datasets, *e.g.*, millions of generated paired images annotated with human preferences. In addition, these human preference datasets can get outdated quickly as the rapid improvements of T2I models lead to higher quality images. In this work, we investigate a *scalable* approach for collecting large-scale and *fully* synthetic datasets for DPO training. Specifically, the preferences for paired images are generated using a pre-trained reward function, eliminating the need for involving humans in the annotation process, greatly improving the

dataset collection efficiency. Moreover, we demonstrate that such datasets allow averaging predictions across multiple models and collecting ranked preferences as opposed to pairwise preferences. Furthermore, we introduce `RankDPO` to enhance DPO-based methods using the ranking feedback. Applying `RankDPO` on SDXL and SD3-Medium models with our synthetically generated preference dataset "`Syn-Pic`" improves both prompt-following (on benchmarks like T2I-Compbench, GenEval, and DPG-Bench) and visual quality (through user studies). This pipeline presents a practical and scalable solution to develop better preference datasets to enhance the performance and safety of text-to-image models.

# 1 INTRODUCTION

While text-to-image (T2I) models (Rombach et al., 2022; Podell et al., 2023; Betker et al., 2023; Esser et al., 2024) have become widespread recently, they still suffer from numerous shortcomings, including challenges with compositional generation (Lin et al., 2024; Wu et al., 2024b), limited ability to render text (Liu et al., 2022a), and lacking of spatial understanding (Chatterjee et al., 2024). There have been several attempts at addressing these issues with larger models (Esser et al., 2024; Ma et al., 2024), improved datasets (Schuhmann et al., 2022; Gadre et al., 2023), and superior language conditioning (Chen et al., 2024b; Pernias et al., 2024). However, these approaches typically involve training larger models from scratch and are not applicable to existing models.

On the other hand, drawing inspiration from Large Language Models (LLMs), aligning T2I models with human feedback has become an important and practical topic to enhance existing T2I models (Liu et al., 2024a). There are two major efforts in this area, namely, 1) collecting large amounts of user preferences images for training (Lee et al., 2023b; Dai et al., 2023; Liang et al., 2024a) and 2) fine-tuning with T2I models with reward functions (Wu et al., 2023b;a; Xu et al., 2023; Kirstain et al., 2023; Zhang et al., 2024a). The *former direction* shows promising results when utilizing Direct Preference Optimization (DPO) (Rafailov et al., 2023), which was first proposed for aligning LLMs with human feedback, to improve the denoising of the more preferred images as compared to the denoising of the less preferred images (Wallace et al., 2024; Li et al., 2024c; Hong et al., 2024b; Liang et al., 2024b). Nevertheless, the existing process for data collection is expensive and the datasets can be outdated quickly, *e.g.*, Pick-a-Picv2 (Kirstain et al., 2023) costs nearly \$50K (Otani et al., 2023) for collecting $512^2$px generated images, while most recent T2I models generate $1024^2$px images. The *latter direction* fine-tunes the T2I models by maximizing the reward functions with the generated images (Black et al., 2024; Fan et al., 2023; Deng et al., 2024; Zhang et al., 2024b; Chen et al., 2024a; Clark et al., 2024; Xu et al., 2023; Prabhudesai et al., 2023; Li et al., 2024d). However, this process is computationally expensive due to the backpropagation through the diffusion process. Additionally, these methods suffer from "reward hacking", where this optimization process increases the reward scores without improving the quality of the generated images.

In this work, we address the above challenges and propose a *scalable* and *cost-effective* solution for aligning T2I models. Specifically, we investigate the efficacy of using *synthetically* labeled preferences in fine-tuning T2I models with DPO-based techniques. While this has been studied in depth in the context of LLMs (Lee et al., 2023a; Bai et al., 2022b), there have been only preliminary explorations in the context of T2I models (Wallace et al., 2024; Wu et al., 2024c). To this end, we introduce the following two novel contributions:

- **Synthetically Labeled Preference Dataset (`Syn-Pic`).** We generate images from different T2I models and label them with multiple pre-trained reward models that can estimate human preference. Therefore, no manual annotation is involved in data collection, making the data collection *cost-effective* and easily *scalable*. By aggregating scores from multiple reward models, we mitigate reward over-optimization (Coste et al., 2024; Eyring et al., 2024). Unlike the conventional pairwise comparisons, we construct a ranking of the generated images for each prompt. While aggregating preferences across multiple human labelers and constructing rankings are possible, these dramatically increase the annotation cost compared to the minimal overhead in our case.
- **Ranking-based Preference Optimization (`RankDPO`).** To leverage the benefits of the richer signal from the rankings, we introduce a ranking-enhanced DPO objective, `RankDPO`, borrowing from the extensive literature on "learning-to-rank" (Burges et al., 2006; Wang et al., 2013; 2018;

Liu et al., 2024c; Song et al., 2024). It weighs the preference loss with discounted cumulative gains (DCG), enabling alignment with the preferred rankings.

We conduct extensive evaluation to demonstrate the advantages of the proposed contributions:

- First, using the same prompts as Pick-a-Picv2 leads to dramatic improvements for the SDXL and SD3-Medium models. We show the improved results on various benchmark datasets, including GenEval (Ghosh et al., 2023) (Tab. 1), T2I-Compbench (Huang et al., 2023) (Tab. 2), and DPG-Bench (Hu et al., 2024) (Tab. 3), as well as, the visual comparisons (examples in Fig. 1) through user studies (Fig. 3).
- Second, we achieve the state-of-the-art results compared to the existing methods on preference optimization, *e.g.*, Tab. 3. More importantly, such results are obtained by only requiring $3\times$ fewer images than Pick-a-Picv2, *i.e.*, Tab. 9.
- Third, even though SD3-Medium (2B parameters) has already been optimized with 3M human preferences through DPO, we are still able to further get significant improvements with our Syn-Pic dataset of 240K images, *e.g.*, Tabs. 1,2,3.

## 2 RELATED WORK

**Text-to-Image Models.** Early works employing GANs for text-to-image synthesis (Reed et al., 2016; Zhang et al., 2017) evolved more recently around diffusion (Sohl-Dickstein et al., 2015; Ho et al., 2020) and rectified flow (Liu et al., 2022b; Lipman et al., 2023; Albergo & Vanden-Eijnden, 2023) models for image and video generation. Following the success of the Stable Diffusion models (Rombach et al., 2022; Podell et al., 2023), several improvements have been proposed, including the use of superior U-Net/transformer backbones (Peebles & Xie, 2023; Bao et al., 2023), stronger language conditioning through superior text encoders (Raffel et al., 2020; Chen et al., 2024c;b) and improved captions Betker et al. (2023); Esser et al. (2024); Chatterjee et al. (2024). In this work, we explore the efficacy of synthetically generated preferences to enhance pre-trained text-to-image model using methods based on learning from human/AI feedback.

**Learning from Human Preferences.** In LLMs, alignment with human preferences (Griffith et al., 2013; Christiano et al., 2017; Bai et al., 2022a) has been crucial in developing chatbots and language assistants. The paradigm of Reinforcement Learning from Human Feedback (RLHF) involved collecting large amounts of user preferences for various prompt output pairs. Following this, reward models were trained to mimic user preferences, after which reinforcement learning algorithms (*e.g.*, PPO (Schulman et al., 2017), REINFORCE (Williams, 1992; Ahmadian et al., 2024)) were used to optimize language models to maximize reward model scores. However, Direct Preference Optimization (Rafailov et al., 2023) along with similar variants (Azar et al., 2024; Ethayarajh et al., 2024; Hong et al., 2024a; Meng et al., 2024; Liu et al., 2024c) emerged as a strong alternative, introducing an equivalent mathematical formulation that enabled training language models directly on user preferences without requiring reward models or reinforcement learning. These insights have since been used more generally in image/video generation (Wallace et al., 2024; Li et al., 2024c). In contrast, we demonstrate the superior efficacy of improving text-to-image models purely from AI feedback, similar to the paradigm of Reinforcement Learning from AI Feedback (Lee et al., 2023a) in LLMs.

**Preference-Tuning of Image Models.** Reward models have been used effectively to fine-tune image generation models using either reinforcement learning (Black et al., 2024; Fan et al., 2023; Deng et al., 2024; Zhang et al., 2024b; Chen et al., 2024a) or reward backpropagation (Lee et al., 2023b; Li et al., 2024d; Prabhudesai et al., 2023; Xu et al., 2023; Clark et al., 2024; Prabhudesai et al., 2024; Domingo-Enrich et al., 2024; Jena et al., 2024). However, this process is computationally expensive and requires additional memory due to backpropagation through the sampling process. Further, it has not yet been successfully applied on larger models at $1024^2$px resolution. As a result, following language modeling literature, DPO techniques have also been adapted to image generation (Wallace et al., 2024; Li et al., 2024c; Liang et al., 2024b; Hong et al., 2024b; Gu et al., 2024), thereby avoiding the expensive training objective. There have also been several methods specifically tailored to improve prompt following in specialized settings (Li et al., 2024b; Hu et al., 2024; Jiang et al., 2024; Sun et al., 2023; Liao et al., 2024). Differently, we demonstrate the possibility of using reward model feedback through the denoising/preference optimization objective as a more general and effective solution than existing approaches for aligning text-to-image models.

## 3 METHOD

In this section, we first provide an overview of diffusion models for text-to-image generation and direct preference optimization for these models. Next, we discuss the process of curating and labeling a scalable preference optimization dataset. Finally, we describe our ranking enabled preference optimization method, called `RankDPO`, to leverage this ranked preference dataset. We describe these two components with illustration in Fig. 2. We provide pseudo-code to train `RankDPO` on `Syn-Pic` in Algorithm 3 in Appendix A.8.

**Notation.** We use the symbol $\mathbf{x} \sim p_{\text{data}}$ to denote the real data drawn from the distribution $p_{\text{data}}$. In our setup, a diffusion process transforms the real image $\mathbf{x}$ to Gaussian noise $\boldsymbol{\epsilon} \sim \mathcal{N}(\mathbf{0}, \mathbf{I})$ with a pre-defined signal-noise schedule $\{\alpha_t, \sigma_t\}_{t=1}^T$. The diffusion model reverses this process by learning a denoiser $\hat{\boldsymbol{\epsilon}}_{\boldsymbol{\theta}}$, a neural network parameterized by $\boldsymbol{\theta}$ to estimate the conditional distribution $p_\theta(\mathbf{x}|\boldsymbol{c})$, where $\boldsymbol{c}$ is the conditioning signal that guides the generation towards the condition. For text-to-image models, we use $\boldsymbol{c}$ as the embedding corresponding to the text-prompt. For brevity, we interchangeably use the symbol $\boldsymbol{c}$ to mean both the text-prompt/embedding.

**Diffusion Models.** Denoising Diffusion Probabilistic Models (DDPMs) learn to predict real data $\mathbf{x} \sim p_{\text{data}}$ by reversing the ODE flow. Specifically, with a pre-defined signal-noise schedule $\{\alpha_t, \sigma_t\}_{t=1}^T$, it samples a noise $\boldsymbol{\epsilon} \sim \mathcal{N}(\mathbf{0}, \mathbf{I})$, and constructs a noisy sample $\mathbf{x}_t$ at time $t$ as $\mathbf{x}_t = \alpha_t \mathbf{x} + \sigma_t \boldsymbol{\epsilon}$. The denoising model $\boldsymbol{\epsilon}_{\boldsymbol{\theta}}$ parameterized by $\boldsymbol{\theta}$ is trained with the objective as:

$$\min_{\boldsymbol{\theta}} \mathbb{E}_{t\sim[1,T], \boldsymbol{\epsilon}\sim\mathcal{N}(\mathbf{0},\mathbf{I})} \|\boldsymbol{\epsilon} - \boldsymbol{\epsilon}_{\boldsymbol{\theta}}(\mathbf{x}_t, \boldsymbol{c})\|^2, \tag{1}$$

where $T$ is the total number of steps, and $\boldsymbol{c}$ is the condition signal.

### 3.1 DPO FOR DIFFUSION MODELS

The Bradley-Terry (BT) model (Bradley & Terry, 1952) defines pairwise preferences with the following formulation:

$$p_{\text{BT}}(\mathbf{x}^w \succ \mathbf{x}^l | \boldsymbol{c}) = \sigma(r(\boldsymbol{c}, \mathbf{x}^w) - r(\boldsymbol{c}, \mathbf{x}^l)), \tag{2}$$

where $\sigma(\cdot)$ is the sigmoid function, $\mathbf{x}^w$ is the more preferred image, $\mathbf{x}^l$ is the less preferred image, and $r(\boldsymbol{c}, \mathbf{x})$ is the reward model that computes alignment score between condition $\boldsymbol{c}$ and image $\mathbf{x}$. Using this preference model, we wish to maximize the following reward objective for a model $\pi$

$$\mathbb{E}_{\tau\sim\pi}[r(\tau)] - \beta \text{KL}(\pi \| \pi_{\text{ref}}) \tag{3}$$

where the KL-regularization term is used to prevent the collapse of the model $\pi$ and $\beta$ controls the strength of the regularization. The regularization ensures that the model being trained ($\pi$), does not deviate too much from the original model $\pi_{\text{ref}}$. From this, Rafailov et al. (2023) demonstrate that the following objective is equivalent to the process of explicit reinforcement learning (*e.g.*, PPO/REINFORCE) with the reward model $r$:

$$L_{\text{DPO}}(\boldsymbol{\theta}) = -\mathbb{E}_{\boldsymbol{c}, \mathbf{x}^w, \mathbf{x}^l} \left[ \log \sigma \left( \beta \log \frac{p_\theta(\mathbf{x}^w|\boldsymbol{c})}{p_{\text{ref}}(\mathbf{x}^w|\boldsymbol{c})} - \beta \log \frac{p_\theta(\mathbf{x}^l|\boldsymbol{c})}{p_{\text{ref}}(\mathbf{x}^l|\boldsymbol{c})} \right) \right], \tag{4}$$

where $p_{\text{ref}}(\mathbf{x}|\boldsymbol{c})$ is the base reference distribution, and $\beta$ controls the distributional deviation.

However, in the context of diffusion models, it is not feasible to compute the likelihood of an image (*i.e.*, $p(\mathbf{x}|\boldsymbol{c})$). Therefore, Wallace et al. (2024) propose a tractable alternative which they prove is equivalent up to a minor relaxation of the original DPO objective.

Given a sample $(\boldsymbol{c}, \mathbf{x}^w, \mathbf{x}^l)$, denoising and reference models ($\boldsymbol{\epsilon}_{\boldsymbol{\theta}}, \boldsymbol{\epsilon}_{\text{ref}}$), we define score function as

$$\mathbf{s}(\mathbf{x}^*, \boldsymbol{c}, t, \boldsymbol{\theta}) = \|\boldsymbol{\epsilon}^* - \boldsymbol{\epsilon}_\theta(\mathbf{x}_t^*, \boldsymbol{c})\|_2^2 - \|\boldsymbol{\epsilon}^* - \boldsymbol{\epsilon}_{\text{ref}}(\mathbf{x}_t^*, \boldsymbol{c})\|_2^2,$$

where $\mathbf{x}_t^* = \alpha_t \mathbf{x}^* + \sigma_t \boldsymbol{\epsilon}^*, \boldsymbol{\epsilon}^* \sim \mathcal{N}(0, I)$ is a noisy latent for input $\mathbf{x}^*$ at time $t$. With this, the updated DPO objective can be defined as follows:

$$\mathcal{L}(\boldsymbol{\theta}) = -\mathbb{E}_{(\boldsymbol{c}, \mathbf{x}^w, \mathbf{x}^l)\sim\mathcal{D}, t\sim[0,T]} \log \sigma \left( -\beta \left( \mathbf{s}(\mathbf{x}^w, \boldsymbol{c}, t, \boldsymbol{\theta}) - \mathbf{s}(\mathbf{x}^l, \boldsymbol{c}, t, \boldsymbol{\theta}) \right) \right). \tag{5}$$

In practice, we randomly sample a timestep ($t$) and compute the denoising objective at this timestep for both the winning ($\mathbf{x}_t^w$) and the losing ($\mathbf{x}_t^l$) sample for both the trainable and the reference models. The DPO objective ensures that for a given conditioning signal $\boldsymbol{c}$, the denosing improves for the winning sample, along with worsening the denoising for the losing sample. This biases the model towards generating images more similar to the preferred images for the condition $\boldsymbol{c}$.

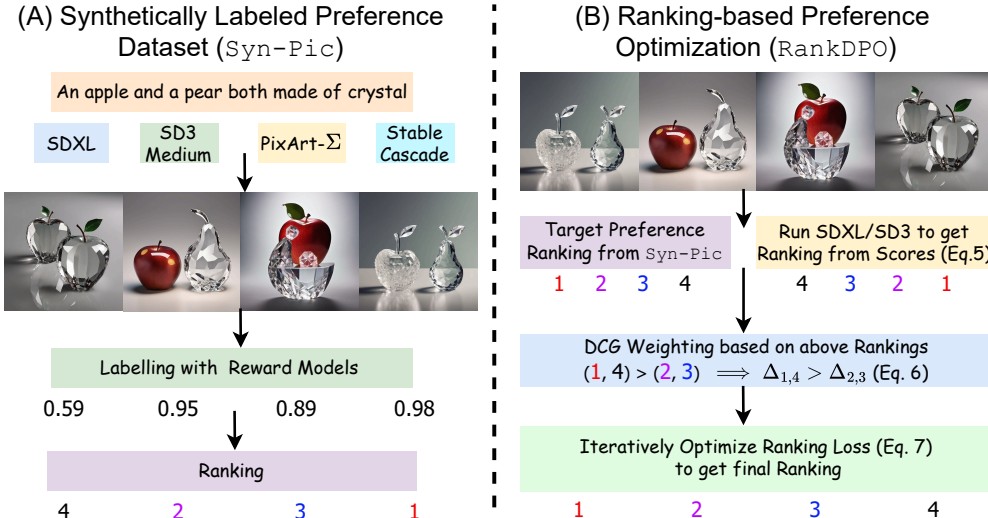

Figure 2: Overview of our two novel components: (A) `Syn-Pic` and (B) `RankDPO`. *Left* illustrates the pipeline to generate a synthetically ranked preference dataset. It starts by collecting prompts and generating images using the same prompt for different T2I models. Next, we calculate the overall preference score using Reward models (*e.g.*, PickScore, ImageReward). Finally, we rank these images in the decreasing order of preference scores. *Right*: Given true preference rankings for generated images per prompt, we first obtain predicted ranking by current model checkpoint using scores $\mathbf{s}_i$ (see Eq. 6). In this instance, although the predicted ranking is inverse of the true rankings, the ranks $(1,4)$ obtains a larger penalty than the ranks $(2,3)$. This penalty is added to our ranking loss through DCG weights (see Eq. 7). Thus, by optimizing $\boldsymbol{\theta}$ with Ranking Loss (see Eq. 8), the updated model addresses the incorrect rankings $(1,4)$. This procedure is repeated over the training process, where the rankings induced by the model aligns with the labelled preferences.

## 3.2 Synthetically Labeled Preference Dataset (Syn-Pic)

In this section, we describe an efficient and scalable method to collect preference dataset $\mathcal{D}$ used in the DPO objective (Eq. 5). Given a list of $N$ text-prompts $\{\boldsymbol{c}_i\}_{i=1}^N$, $\mathcal{D}$ consists of paired preferences which denote winning and losing images, *i.e.*, $\{\boldsymbol{c}_i, \mathbf{x}_i^w, \mathbf{x}_i^l\}_{i=1}^N$. Pick-a-Picv2 (Kirstain et al., 2023) is an example of a preference dataset used in earlier works, consisting of nearly 58K prompts and 0.85M preference pairs. Traditionally, the data collection process involves human annotations of images generated by text-to-image models, which is expensive due to human labeling costs. Further, these hand-curated datasets become outdated quickly due to improvements in text-to-image models.

We collect a new preference dataset by generating images from various state-of-the-art T2I models (*e.g.*, SD3-Medium (Esser et al., 2024), StableCascade (Pernias et al., 2024), Pixart-$\Sigma$ (Chen et al., 2024b), and SDXL (Podell et al., 2023)) for the same prompts as the Pick-a-Picv2 dataset. Further, we eliminate the human annotation cost by labeling these samples using existing off-the-shelf human-preference models, *e.g.*, HPSv2.1 (Wu et al., 2023a). However, different reward models may have complementary strengths (*e.g.*, some focus more on visual quality, others are better at text-image alignment, etc.). Therefore, we propose to aggregate the preferences from 5 different models, including HPSv2.1 (Wu et al., 2023a), MPS (Zhang et al., 2024a), PickScore (Kirstain et al., 2023), VQAScore (Lin et al., 2024), and ImageReward (Xu et al., 2023). For each prompt $\boldsymbol{c}$ and image $\mathbf{x}_i^k$, we compute the probability of the an image being preferred over other images over all rewards. by aggregating the total number of wins compared to the total number of comparisons for $\mathbf{x}_i^k$. This score $\phi(\mathbf{x}_i^k)$ is used to rank the generated images in the decreasing order of aggregate scores, resulting in target preference ranking, and is also used for the gain function in Sec. 3.3. Thus, for $k$ T2I models, we obtain $\mathcal{D} = \{\boldsymbol{c}_i, \mathbf{x}_i^1, \mathbf{x}_i^2, \ldots, \mathbf{x}_i^k, \phi(\mathbf{x}_i^1), \phi(\mathbf{x}_i^2), \ldots, \phi(\mathbf{x}_i^k)\}_{i=1}^N$, a fully synthetically ranked preference dataset. We describe it in a detailed procedure in Algorithm 1 in Appendix A.8.

**Discussion.** Our data collection method has several benefits as highlighted below.

- **Cost Efficiency.** We can generate arbitrarily large preference dataset, since there is no human in the annotation loop, both image-generation and labelling is done using off-the-shelf models, reducing the dataset curation cost. For instance, it requires $\approx \$50K$ to collect Pick-a-Picv2 (Kirstain et al., 2023) dataset, in contrast, we can collect a similar scale dataset with $\approx \$200$.

- **Scalability.** With reduced dataset collection cost, we can iterate over new text-to-image models, removing the issue of older preference datasets becoming obsolete with new models.
- **Ranking-based Preference Optimization.** Since we run multiple T2I models per prompt, we collect a ranked preference list compared to just paired data in earlier datasets. This enables us to explore ranking objective in the preference optimization. We explore this objective in next section.

### 3.3 RANKING-BASED PREFERENCE OPTIMIZATION (RANKDPO)

Unlike the DPO objective which focuses on pairwise preferences, our synthetic dataset generates multiple images per prompt, resulting in a ranked preference dataset. Therefore, we would like to optimize the many-way preference at once instead of purely relying on pairwise preferences. Specifically, given a text-prompt $c$, and generated images in a ranked order of preference $\{\mathbf{x}^1, \mathbf{x}^2, \ldots, \mathbf{x}^k\}$, we want to ensure that the denoising for image $\mathbf{x}^i$ is better than $\mathbf{x}^j$ for all $i > j$. To enforce this, we draw inspiration from the Learning to Rank (LTR) literature, and re-purpose the LambdaLoss (Wang et al., 2018) by adding the DCG weights to each sample, similar to Liu et al. (2024c) as follows:

Given a sample $(c, \{\mathbf{x}^i\}_{i=1}^k, \{\phi(i)\}_{i=1}^k)$, denoising, reference models $(\epsilon_\theta, \epsilon_{\text{ref}})$, we define score as

$$\mathbf{s}_i \triangleq \mathbf{s}(\mathbf{x}^i, c, t, \boldsymbol{\theta}) = \|\boldsymbol{\epsilon}^i - \boldsymbol{\epsilon}_\theta(\mathbf{x}_t^i, c)\|_2^2 - \|\boldsymbol{\epsilon}^i - \boldsymbol{\epsilon}_{\text{ref}}(\mathbf{x}_t^i, c)\|_2^2, \qquad (6)$$

where $\mathbf{x}_t^i = \alpha_t \mathbf{x}^i + \sigma_t \boldsymbol{\epsilon}^i, \boldsymbol{\epsilon}^i \sim \mathcal{N}(0, I)$ is a noisy latent for input $\mathbf{x}^i$ at time $t$. This score measures how much better or worse the model prediction is compared to the reference model for the given condition $c$.

After computing the scores $\mathbf{s}_i$, the images are ranked from the most preferred (lowest $\mathbf{s}_i$) to the least preferred (highest $\mathbf{s}_i$). This is the predicted rank for $\{\mathbf{x}^i\}_{i=1}^k$ using model $\theta$. We use the ground truth scores $\phi(i)$ to obtain the true preference ranking $\tau(\cdot)$. The rank of each image $\mathbf{x}^i$ is denoted by $\tau(i)$, where $\tau(i) = 1$ for the best image, $\tau(i) = 2$ for the second best, and so on. The gain for each sample $\phi(i)$ is the average probability that sample $i$ is preferred over all other samples $j$ according to human preference reward model scoring.

Using $\tau(i)$ and $\phi(i)$, we define the gain function $G_i$ and the discount function $D(\tau(i))$ as:

$$G_i = 2^{\phi(i)} - 1; \quad D(\tau(i)) = \log(1 + \tau(i)).$$

The discount function decreases as the rank $\tau(i)$ increases, ensuring that higher-ranked images (those with a lower $\tau(i)$) have a greater influence on the final loss. The logarithmic form of the discount function smooths out the penalty differences between consecutive ranks, making the model more robust to small ranking errors, especially for lower-ranked images.

We define the weight between two image pairs $(\mathbf{x}^i, \mathbf{x}^j)$ as

$$\Delta_{i,j} = |G_i - G_j| \cdot \left| \frac{1}{D(\tau(i))} - \frac{1}{D(\tau(j))} \right|. \qquad (7)$$

Finally, putting all these together, the RankDPO loss is then formulated as:

$$\mathcal{L}_{\text{RankDPO}}(\boldsymbol{\theta}) = -\mathbb{E}_{(c, \mathbf{x}^1, \mathbf{x}^2, \ldots, \mathbf{x}^k) \sim \mathcal{D}, \, t \sim [0, T]} \left[ \sum_{i > j} \Delta_{i,j} \log \sigma \left( -\beta \left( \mathbf{s}(\mathbf{x}^i, c, t, \boldsymbol{\theta}) - \mathbf{s}(\mathbf{x}^j, c, t, \boldsymbol{\theta}) \right) \right) \right], \qquad (8)$$

where $\sigma(\cdot)$ is the sigmoid function and $\beta$ controls the strength of the KL regularization. We describe the training process in a detailed procedure in Algorithm 2 in Appendix A.8.

This loss function encourages the model to produce images that not only satisfy pairwise preferences, but also respect the overall ranking of images generated for the same prompt. By weighting the traditional DPO objective with gains and discounts derived from the ranking, we ensure that the model prioritizes the generation of higher-quality images according to the ranking, leading to more consistent improvements in both aesthetics and prompt alignment.

## 4 EXPERIMENTS

**Implementation Details.** We perform our experiments using the open-source SDXL (Podell et al., 2023) and SD3-Medium models (Esser et al., 2024). We use 58K prompts from Pick-a-Picv2 and four models, *i.e.*, SDXL, SD3-Medium, Pixart-$\Sigma$, and Stable Cascade, to prepare Syn-Pic. We train RankDPO with 8 A100 GPUs for 16 hours with a batch size of 1024 trained for 400 steps. Further details about the training and evaluation metrics are provided in Appendix A.4.

Table 1: **Quantitative Results on GenEval**. `RankDPO` improves results on most categories, notably "two objects", "counting", and "color attribution" for SDXL and SD3-Medium.

| Model | Mean ↑ | Single ↑ | Two ↑ | Counting ↑ | Colors ↑ | Position ↑ | Color Attribution ↑ |
|---|---|---|---|---|---|---|---|
| SD v2.1 | 0.50 | 0.98 | 0.51 | 0.44 | 0.85 | 0.07 | 0.17 |
| PixArt-$\alpha$ | 0.48 | 0.98 | 0.50 | 0.44 | 0.80 | 0.08 | 0.07 |
| PixArt-$\Sigma$ | 0.53 | 0.99 | 0.65 | 0.46 | 0.82 | 0.12 | 0.12 |
| Stable Cascade | 0.53 | 1.00 | 0.61 | 0.49 | 0.86 | 0.08 | 0.13 |
| DALL-E 2 | 0.52 | 0.94 | 0.66 | 0.49 | 0.77 | 0.10 | 0.19 |
| DALL-E 3 | 0.67 | 0.96 | 0.87 | 0.47 | 0.83 | 0.43 | 0.45 |
| SDXL | 0.55 | 0.98 | 0.74 | 0.39 | 0.85 | 0.15 | 0.23 |
| **SDXL (Ours)** | **0.61** | 1.00 | 0.86 | 0.46 | 0.90 | 0.14 | 0.29 |
| SD3-Medium | 0.70 | 1.00 | 0.87 | 0.63 | 0.84 | 0.28 | 0.58 |
| **SD3-Medium (Ours)** | **0.74** | 1.00 | 0.90 | 0.72 | 0.87 | 0.31 | 0.66 |

Table 2: **Quantitative Results on T2I-CompBench**. `RankDPO` provides consistent improvements on all categories for both SDXL and SD3-Medium.

| Model | Attribute Binding | | | Object Relationship | | Complex↑ |
|---|---|---|---|---|---|---|
| | Color ↑ | Shape↑ | Texture↑ | Spatial↑ | Non-Spatial↑ | |
| SD1.4 | 37.65 | 35.76 | 41.56 | 12.46 | 30.79 | 30.80 |
| PixArt-$\alpha$ | 68.86 | 55.82 | 70.44 | 20.82 | 31.79 | 41.17 |
| PixArt-$\Sigma$ | 57.28 | 45.61 | 62.48 | 28.23 | 31.17 | 44.71 |
| Stable Cascade | 48.62 | 40.18 | 54.96 | 24.60 | 31.09 | 40.57 |
| DALL-E 2 | 57.50 | 54.64 | 63.74 | 12.83 | 30.43 | 36.96 |
| SDXL | 58.79 | 46.87 | 52.99 | 21.31 | 31.19 | 32.37 |
| **SDXL (Ours)** | 72.33 | 56.93 | 69.67 | 24.53 | 31.33 | 45.47 |
| SD3-Medium | 81.31 | 59.06 | 75.91 | 34.30 | 31.13 | 47.93 |
| **SD3-Medium (Ours)** | 83.26 | 63.45 | 78.72 | 36.49 | 31.25 | 48.65 |

## 4.1 COMPARISON RESULTS

**Short Prompts.** In Tab. 1, we report results on GenEval (Ghosh et al., 2023). `RankDPO` consistently improves the performance on almost every category, leading to an averaged performance gain from 0.55 to 0.61 for SDXL and from 0.70 to 0.74 for SD3-Medium. In particular, we observe large improvements on "two objects", "counting" and "color attribution", where there are gains of nearly 10%.We observe a similar trend on T2I-Compbench (Huang et al., 2023) in Tab. 2, where SDXL gains by over 10% on "Color" and "Texture" and achieves improvements in other categories.

**Long Prompts.** In Tab. 3, we further evaluate models for visual quality and prompt alignment on DPG-Bench (Hu et al., 2024), which consists of long and detailed prompts. To measure prompt alignment, we employ both the original DSG metric (Cho et al., 2024) and VQAScore (Lin et al., 2024), while for visual quality, we use the Q-Align model (Wu et al., 2024a). We notice that Diffusion-DPO (denoted as DPO-SDXL) (Wallace et al., 2024) trained on Pick-a-Picv2 is able to provide meaningful improvements on prompt alignment, while fine-tuning SDXL with MaPO (Hong et al., 2024b) and SPO (Liang et al., 2024b) (denoted as MaPO-SDXL and SPO-SDXL) improves visual quality. However, `RankDPO`, despite being trained only on synthetic preferences, improves *all* metrics by *significant* amounts (*e.g.*, 74.51 to 79.26 on DSG score and 0.72 to 0.81 on Q-Align score for SDXL) and achieves the *state-of-the-art* prompt alignment metrics. For SD3-Medium, we continue to see improved model performance after fine-tuning with our proposed `RankDPO`.

**User Study.** To further validate the effectiveness of our approach, we perform a user study on 450 prompts from DPG-bench. We ask users to choose the better image based on their overall preference, *i.e.*, combining text-image alignment and visual quality. Fig. 3 shows that `RankDPO` has a superior win-rate compared to both DPO-SDXL (Wallace et al., 2024) and SDXL (Podell et al., 2023), indicating the efficacy in enhancing the overall quality of the generated images.

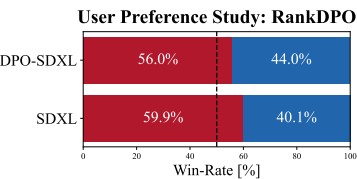

Figure 3: Win rates of our approach on human evaluation.

Table 3: Quantitative results on DPG-Bench. DSG (Cho et al., 2024) and VQAScore (Lin et al., 2024) measure prompt following using VQA models while Q-Align (Wu et al., 2024a) measures visual quality using multimodal LLMs.

| Model Name | Prompt Alignment | | Visual Quality |
|---|---|---|---|
| | DSG Score | VQA Score | Q-Align Score |
| SD1.5 | 63.18 | - | - |
| SD2.1 | 68.09 | - | - |
| Pixart-$\alpha$ | 71.11 | - | - |
| Playgroundv2 | 74.54 | - | - |
| Pixart-$\Sigma$ | 80.54 | - | - |
| Stable Cascade | 70.92 | - | - |
| DALL-E 3 | 83.50 | - | - |
| SDXL | 74.65 | 84.33 | 0.72 |
| DPO-SDXL | 76.74 | 85.67 | 0.74 |
| MaPO-SDXL | 74.53 | 84.54 | 0.80 |
| SPO-SDXL | 74.73 | 84.71 | 0.82 |
| **SDXL (Ours)** | **79.26** | **87.52** | **0.81** |
| SD3-Medium | 85.54 | 90.58 | 0.67 |
| **SD3-Medium (Ours)** | **86.78** | **90.99** | **0.68** |

Table 4: Effect of the preference labelling and data quality on the final model. We see `Syn-Pic` is able to consistently improve performance along with `RankDPO`.

| Model Name | Prompt Alignment | | Visual Quality |
|---|---|---|---|
| | DSG Score | VQA Score | Q-Align Score |
| SDXL | 74.65 | 84.33 | 0.72 |
| DPO (Random Labelling) | 75.66 | 84.42 | 0.74 |
| DPO (HPSv2) | 78.04 | 86.22 | 0.83 |
| DPO (Pick-a-Picv2) | 76.74 | 85.67 | 0.74 |
| DPO (5 Rewards) | 78.84 | 86.27 | 0.81 |
| RankDPO (Only SDXL) | 78.40 | 86.76 | 0.74 |
| RankDPO | 79.26 | 87.52 | 0.81 |

Table 5: Analysis of the learning objectives. While DPO improves over fine-tuning, `RankDPO` provides further gains.

| Model Name | Prompt Alignment | | Visual Quality |
|---|---|---|---|
| | DSG Score | VQA Score | Q-Align Score |
| SDXL | 74.65 | 84.33 | 0.72 |
| Supervised Fine-Tuning | 76.56 | 85.45 | 0.78 |
| Weighted Fine-Tuning | 77.02 | 85.55 | 0.79 |
| DPO | 78.84 | 86.27 | 0.81 |
| DPO + Gain Weights | 79.15 | 87.43 | 0.82 |
| RankDPO (Ours) | 79.26 | 87.52 | 0.81 |

Table 6: **Evaluating RankDPO vs Diffusion-DPO on GenEval with Syn-Pic dataset**. We train Diffusion-DPO and RankDPO on our proposed Syn-Pic dataset and evaluate their performance on GenEval benchmark. `RankDPO` improves results on most categories, notably "two objects", "counting", and "color attribution" for SDXL and SD3-Medium.

| Model | Mean ↑ | Single ↑ | Two ↑ | Counting ↑ | Colors ↑ | Position ↑ | Color Attribution ↑ |
|---|---|---|---|---|---|---|---|
| SDXL | 0.55 | 0.98 | 0.74 | 0.39 | 0.85 | 0.15 | 0.23 |
| Diffusion-DPO | 0.59 | 0.99 | 0.84 | 0.49 | 0.87 | 0.13 | 0.24 |
| **RankDPO** | **0.61** | **1.00** | **0.86** | 0.46 | **0.90** | 0.14 | **0.29** |
| SD3-Medium | 0.70 | 1.00 | 0.87 | 0.63 | 0.84 | 0.28 | 0.58 |
| Diffusion-DPO | 0.72 | 1.00 | 0.90 | 0.63 | 0.87 | 0.32 | 0.58 |
| **RankDPO** | **0.74** | **1.00** | **0.91** | **0.72** | 0.87 | 0.31 | **0.66** |

**Qualitative Examples** for prompts from DPG-Bench (Hu et al., 2024) are presented in Fig. 4. Compared to the base SDXL and other preference-tuned models, `RankDPO` provides superior prompt following. For instance, we see improved rendering of text, capturing all the objects described in the prompts which are missed by other models, and better modeling of complex relations between objects in the image. To evaluate the fidelity of the generated images, we also measure the FID on MJHQ-30k (Li et al., 2024a) with SDXL in Tab. 10 and demonstrate consistent improvements.

**Discussion of Computation Cost.** We require 10 A100 GPU days to generate images and label the preferences, which is a one-time cost. Running `RankDPO` for 400 steps on the generated data takes about 6 GPU days for SDXL at $1024^2$px. In contrast, existing reward optimization methods (Li et al., 2024d; Zhang et al., 2024b) take 64-95 A100 GPU days with the smaller SD1.5 model at $512^2$px. Similarly, compared to Diffusion-DPO (Wallace et al., 2024), `RankDPO` trains on one-third of the data while avoiding manually curated preferences. There are also methods enhancing text-to-image models by using text encoders such as T5/LLaMA models (Hu et al., 2024; Liu et al., 2024b), which require 10M to 34M densely captioned images and train for 50-120 A100 GPU days.

## 4.2 ABLATION ANALYSIS

**Effect of Data and Labelling Function.** Since generating the preferences is a crucial aspect of `RankDPO`, we evaluate different labelling choices in Tab. 4. We experiment with random labelling where preferences are randomly chosen and apply DPO. This is able to only provide minimal improvements in performance (74.65 to 75.66 DSG score). We also show the results with pairwise preferences from a single reward model (HPSv2.1) and averaging preferences from 5 models. While HPSv2.1 provides good improvements for both prompt alignment and visual quality, ensembling the predictions across multiple models improves the results further. We also note that these results outperform DPO applied on Pick-a-Picv2, highlighting the importance of the image quality while

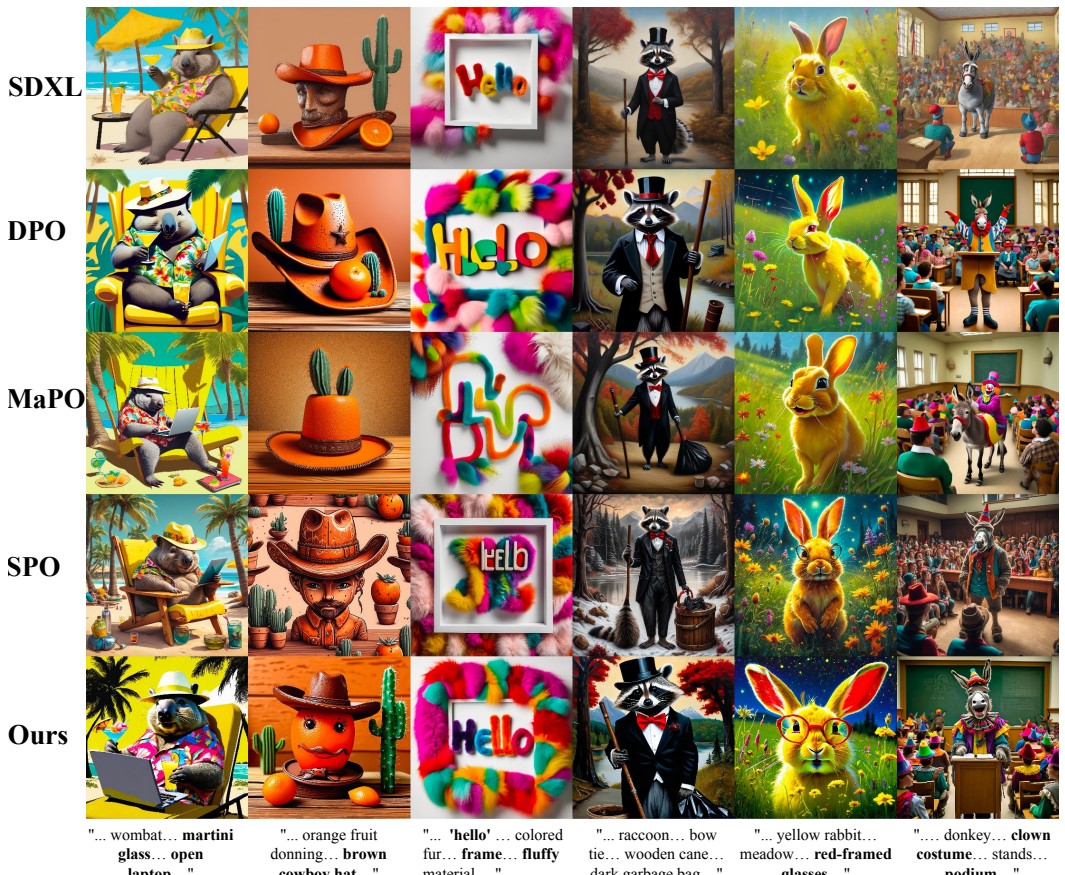

SDXL

DPO

MaPO

SPO

Ours

"... wombat... **martini glass**... **open laptop**..."
"... orange fruit donning... **brown cowboy hat**..."
"... **'hello'** ... colored fur... **frame**... **fluffy** material ..."
"... raccoon... bow tie... wooden cane... dark garbage bag..."
"... yellow rabbit... meadow... **red-framed glasses**..."
".... donkey... **clown costume**... stands... **podium**..."

Figure 4: Comparison among different preference optimization methods (DPO, SPO, and MaPO) and `RankDPO` for SDXL. The results illustrate that our method generates images with better prompt alignment, text rendering, and aesthetic quality.

constructing preference datasets. Finally, we investigate the impact of the different models used to construct `Syn-Pic`. This is done by constructing a similar dataset with SDXL images by only varying the seed. While we nearly get the same improvements in prompt alignment, we only see a small improvement in visual quality. This indicates that synthetic preference-tuning can be applied to any model on its outputs, however, having images from different models can further improve results.

**Analysis of Learning Objective.** A critical aspect of preference optimization is the choice of learning objectives and we perform various experiments in Tab. 5 to compare them. Besides the regular DPO formulation, several works show the benefits of supervised fine-tuning on curated high-quality data (Dai et al., 2023; Liang et al., 2024a; Li et al., 2024a), which also we take into the comparisons. The baseline includes the following:

- *Supervised Fine-Tuning* that subselects the winning image from each pairwise comparison and fine-tunes SDXL on this subset.
- *Weighted Fine-Tuning* that fine-tunes SDXL on all the samples, but assigns a weight to each sample based on the HPSv2.1 scores (Wu et al., 2023a), similar to Lee et al. (2023b).
- *DPO + Gain Function Weighting.* The DPO objective can be improved by incorporating the reward information: by weighting the samples using the gain function.

We can see that the best results are achieved by `RankDPO`, highlighting the benefits of incorporating ranking criteria based-on paired preferences to strengthen preference optimization. We further evaluate the efficacy of `RankDPO` as compared to the standard DPO objective on GenEval in Tab. 6. We demonstrate that for both SDXL and SD3-Medium, DPO provides improvements over the base model, while `RankDPO` is able to provide consistent improvements over the DPO objective.

## 5 CONCLUSION AND DISCUSSION

In this work, we introduce a powerful and cost-effective recipe for the preference optimization of text-to-image models. In particular, we demonstrate how synthetically generating a preference optimization dataset can enable the collection of superior signals (*e.g.*, rankings *vs.* pairwise preferences and ensembling preferences across models). We also introduce a simple method to leverage the stronger signals, leading to state-of-the-art results on various benchmarks for prompt following and visual quality for both the diffusion and rectified flow models. We hope our work paves the way for future work on scaling effective post-training solutions for text-to-image models.

**Limitations.** We rely solely on the prompts from Pick-a-Picv2 (Kirstain et al., 2023) for constructing our preference dataset. While this allows us to fairly compare to prior work on preference optimization, our dataset is limited by the quantity and diversity of the prompts in Pick-a-Picv2. Expanding the prompts to include different use cases would significantly enhance the utility of the dataset and improve the quality of the downstream models. Additionally, we focus only on text-image alignment and visual quality. However, preference optimization is also well-suited to improving the safety of the text-to-image models, which can also be investigated in future work. Finally, we also reply purely on off-the-shelf reward models. Some of these models have shown impressive performance in recent times, even outperforming a single user on several benchmarks. However, the annotations from these reward models could still have problems, and stronger reward models in the future would be crucial in strengthening the results.

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

# A APPENDIX

## A.1 COMPARISON TO OTHER METHODS

We also investigate methods like ELLA (Hu et al., 2024) which replace the CLIP text encoder in SDXL with a LLM based text-encoder (*e.g.*, T5-XL) and use an adapter (470M params in the case of ELLA) to project these features to the original feature space. While both ELLA and `RankDPO` achieve similar performance on T2I-Compbench and DPG-bench as in Tab. 7, we must note that ELLA takes $18\times$ the training time and over $100\times$ images. Moreover, this imposes the additional cost of including the T5/LLaMa model and using the timestep based adapter (470M params) at every timestep, leading to increased inference time. We also compare ELLA and other preference optimization methods in Tab. 8. We see that `RankDPO` trained on `Syn-Pic` provides the best trade-off in terms of training data requirements, computational resources (training time) and downstream performance (as measured with the DPG-bench score). Finally, we do not have comparisons against methods that perform reward fine-tuning, since they need 100 A100 days to be applied at a large-scale for the smaller SD1.5 model at 512 resolution and have not been applied sucessfully to the larger SDXL model at 1024 resolution or show minimal benefits in enhancing text-image alignment (Jena et al., 2024).

Table 7: Comparison of T2I-Compbench Dataset with DPG-Bench, including model attributes, training time, and inference time increases.

| Dataset | Color | Shape | Texture | Spatial | Non-Spatial | DPG Score | Train Time (A100 Days) | Training Data | Same Inference Time |
|---|---|---|---|---|---|---|---|---|---|
| SDXL | 58.79 | 46.87 | 52.99 | 21.31 | 31.19 | 74.65 | - | - | ✓ |
| ELLA (SDXL) | 72.60 | 56.34 | 66.86 | 22.14 | 30.69 | 80.23 | 112 | 34M | ✗ |
| RankDPO (SDXL) | 72.33 | 56.93 | 69.67 | 24.53 | 31.33 | 79.26 | 6 | 0.24M | ✓ |

Table 8: Comparing features of our proposal against baselines that aim to improve T2I model quality post-training. ELLA* also replaces the CLIP text-encoders with T5-XL text-encoder and a 470M parameter adapter applied at each timestep, thereby increasing the inference cost.

| Method | Training Images | A100 GPU days | Equal Inference Cost | DPG-Bench Score |
|---|---|---|---|---|
| DPO | 1.0M | 30 | ✓ | 76.74 |
| MaPO | 1.0M | 25 | ✓ | 74.53 |
| SPO | - | 5 | ✓ | 74.73 |
| ELLA* | 34M | 112 | ✗ | 80.23 |
| Ours | 0.24M | 6 | ✓ | 79.26 |

## A.2 BINARY CASE OF RANKDPO OBJECTIVE

The binary setting of `RankDPO` ends up with a fixed value for the discount function (since there are only two ranks $1, 2$) and as a result, the only addition is the gain function, which we discuss in Tab. 5.

## A.3 RANKDPO APPLIED TO OTHER DPO OBJECTIVES

A crucial benefit of `RankDPO` is that it can be applied independently on any pairwise objective. More formally, given weight between two image pairs $(\mathbf{x}^i, \mathbf{x}^j)$ as

$$\Delta_{i,j} = |G_i - G_j| \cdot \left| \frac{1}{D(\tau(i))} - \frac{1}{D(\tau(j))} \right|. \tag{9}$$

Then, the generalized objective is written as:

$$\mathcal{L}_{\text{RankGeneralized}}(\boldsymbol{\theta}) = -\mathbb{E}(\boldsymbol{c}, \mathbf{x}^1, \mathbf{x}^2, \ldots, \mathbf{x}^k) \sim \mathcal{D}, , t \sim [0, T] \left[ \sum_{i>j} \Delta_{i,j} \cdot \mathcal{L}_{\text{pairwise}} \left( \mathbf{s}(\mathbf{x}^i, \boldsymbol{c}, t, \boldsymbol{\theta}), \mathbf{s}(\mathbf{x}^j, \boldsymbol{c}, t, \boldsymbol{\theta}) \right) \right],$$

$$\tag{10}$$

Such a formulation would let us extend other preference objectives (Hong et al., 2024b) to include ranking based cues to improve the optimization.

### A.4 DETAILED EXPLANATION OF EVALUATIONS

T2I-Compbench consists of 6000 compositional prompts from 6 different categories (color, shape, texture, spatial, non-spatial, complex). Following the trends of recent protocols (Bakr et al., 2023; Lin et al., 2024), the evaluation for these prompts are done using a combination of VQA models, object detectors and vision-language model scores (*e.g.*, CLIPScore (Hessel et al., 2021)).

GenEval consists of 553 prompts comprising different challenges (single object, two objects, counting, position, color, color attribution). These are mostly evaluated using object detectors.

DPG-Bench aggregates prompts from several sources, and lengthens them using LLMs. These prompts on average have 67 words making it extremely challenging for prompt following. The generated images are mostly evaluated using VQA models under the Davidsonian Scene Graph (Cho et al., 2024) framework. We use the following evaluation metrics for different benchmarks:

- GeneEval. The evaluation for GenEval is performed using the Maskformer (Cheng et al., 2021) object detection models. This is used to determine if the image contains objects specified in the prompts. For color, a CLIP model is used to identify the color of the objects.
- T2I-CompBench: Attribute Binding uses a BLIP-VQA model (Li et al., 2022b) to ask different (upto 8) questions about the generated images, and is used to validate if the answered questions match the details specified in the prompt.
- T2I-CompBench: Spatial uses a Unidet (Zhou et al., 2022) model to perform object detection to see if the objects in the generated images follow the spatial orientation specified in the prompt.
- T2I-CompBench: Non-Spatial computes the CLIPScore for the prompt and the generated image.
- T2I-CompBench: Complex averages the score computed from Attribute Binding, Spatial, and Complex.
- DPG-Bench: DSG uses the Davidsonian Scene Graph (Cho et al., 2024) to compute question answer pairs and use a VQA model (mPLUG) (Li et al., 2022a) to answer the questions before computing the percentage of questions correctly answered.
- DPG-Bench: VQAScore (Lin et al., 2024) trains a multimodal LLM with a CLIP encoder and Flan-T5 decoder to predict the likelihood of the prompt being appropriate for the image.
- DPG-Bench: Q-Align Aesthetic Score (Wu et al., 2024a) finetunes a multimodal LLM (*e.g.*, LLaVA (Liu et al., 2023)) to predict the aesthetic score of an image from a scale of 0 to 1.

### A.5 COST ANALYSIS.

We provide the estimates for the cost of labeling Pick-a-Picv2 as compared to `Syn-Pic` in Tab. 9. Even excluding the cost of generating 2M images, labeling 1̃M pairwise preferences becomes expensive when following standard guidelines of Otani et al. (2023) and paying $0.05 per comparison. However, in contrast, `Syn-Pic` costs $< \$20$ for labeling preferences using *five* different reward models, since each of them need only a few hours on a single GPU to label the preferences. We also note that using an LLM like GPT-4o to generate the comparisons would take over $450 to just process all the images from `Syn-Pic`. Here, the bigger cost is in generating 4 images for the 58K prompts from Pick-a-Picv2, which can still be completed in $< \$200$.

### A.6 MJHQ-30K EVALUATION

We evaluate the FID on MJHQ-30k prompts with the images from Midjourney as reference (Li et al., 2024a) in Tab. 10. We observe consistent improvements over the reported results for SDXL-refiner, indicating that we are able to generate high-fidelity images.

Table 9: Cost comparison of generating and labelling Pick-a-Picv2 *vs.* `Syn-Pic`

| Item | Pick-a-Picv2 | `Syn-Pic` |
|---|---|---|
| Number of prompts | 58 000 | 58 000 |
| Number of images | 1 025 015 | 232 000 |
| Number of preferences | 959 000 | N/A |
| Image generation cost | N/A | $185.60 |
| Annotation/Labelling cost | $47 950.00 | < $20.00 |
| **Total cost** | $47 950.00 | < $205.60 |

Table 10: FID Scores on MJHQ-30k Prompts for 10 Categories and Overall. `RankDPO` consistently outperforms SDXL-Refiner on 9/10 categories.

| Category | Animals | Art | Fashion | Food | Indoor | Landscape | Logo | People | Plants | Vehicles | Overall |
|---|---|---|---|---|---|---|---|---|---|---|---|
| SDXL-Refiner | 28.93 | 31.05 | 28.90 | 30.09 | 28.83 | 30.78 | 36.67 | 35.56 | **28.42** | 24.45 | 9.55 |
| RankDPO SDXL | **24.37** | **27.22** | **22.91** | **25.02** | **26.01** | **28.61** | **29.74** | **27.54** | 30.27 | **21.83** | **7.99** |

## A.7 ADDITIONAL EXAMPLES

We provide further qualitative comparisons against SDXL (Fig. 6) and other preference optimization methods (Fig. 5) from prompts of DPG-Bench. We see improved prompt following: specifically objects mentioned in the prompt which can easily be missed by SDXL are captured by our model. Further, we also see improved modeling on finer details and relations in the generated images. We also provide an example for SD3-Medium in Fig. 7. In addition to the trends observed before, we observe examples where we are able to fix some deformities in the generations of SD3-Medium.

## A.8 PSEUDO CODE

In Sec. 3, we described our two novel components: (a) `Syn-Pic`, and (b) `RankDPO`. Although we provide a method overview in Fig. 2, for completion we also present the detailed workings of these two components in a procedural manner. Algorithm 1 describes the data collection process given the set of prompts, T2I models, and human preference reward models. Algorithm 2 describes the pseudo code to train a diffusion model using `RankDPO`. It takes as input the ranked preference dataset (`Syn-Pic`), reference model $\theta_{\text{ref}}$, initial model $\theta_{\text{init}}$, and other hyper-parameters that control the training and noise-signal schedule in the diffusion process. Finally, Algorithm 3 combines these two procedures to describe our end-to-end data generation and training process.

---

**Algorithm 1** `DataGen`: Generate Synthetically Labeled Ranked Preference Dataset (`Syn-Pic`)

---

**Input:** $N$ prompts ($\mathcal{P} = \{c_i\}_{i=1}^N$), $k$ T2I Models ($\{\theta_i\}_{i=1}^k$), $n$ Reward Models ($\{\mathcal{R}_\psi^i\}_{i=1}^n$)
**Output:** Ranked Preference Dataset $\mathcal{D}$
**Initialize:** Synthetic dataset $\mathcal{D} = \emptyset$
**for** $c$ in $\mathcal{P}$ **do**
    Generate $k$ images $\mathbf{x}^1, \mathbf{x}^2, \dots, \mathbf{x}^k = \theta_1(c), \theta_2(c), \dots, \theta_k(c)$
    Initialize preference counts $C_i = 0; \ \forall i \in \{1, \dots, k\}$
    **for** each reward model $\mathcal{R}_\psi^l$ **do**
        Compute scores $R_i^l = \mathcal{R}_\psi^l(\mathbf{x}^i, c); \ \forall i \in \{1, \dots, k\}$
        **for** each pair $(i, j)$ with $i \neq j$ **do**
            **if** $R_i^l > R_j^l$ **then**
                Increment preference count $C_i = C_i + 1$
    Compute probabilities $\phi(\mathbf{x}^i) = \frac{C_i}{n \cdot (k-1)}; \ \forall i \in \{1, \dots, k\}$
    Store entry $(c, \mathbf{x}^1, \mathbf{x}^2, \dots, \mathbf{x}^k, \phi(\mathbf{x}^1), \phi(\mathbf{x}^2), \dots, \phi(\mathbf{x}^k))$ in $\mathcal{D}$
**return** Ranked Preference Dataset $\mathcal{D}$

---

---

**Algorithm 2** `RankDPO`: Ranking-based Preference Optimization using `Syn-Pic`

---

**Input:** Ranked Preference Dataset $\mathcal{D}$, Initial model $\theta_{\text{init}}$, Reference model $\theta_{\text{ref}}$
**Input:** Pre-defined signal-noise schedule $\{\alpha_t, \sigma_t\}_{t=1}^T$
**Hyper-parameters:** # Optimization Steps $(m)$, Learning Rate $(\eta)$, Divergence Control $\beta$
**Initialize:** $\theta = \theta_{\text{init}}$
**Output:** Fine-tuned model $\theta^{\text{RankDPO}}$
**for** iter $= 0$ **to** $m$ **do**
    Sample entry $(\boldsymbol{c}, \mathbf{x}^1, \mathbf{x}^2, \ldots, \mathbf{x}^k, \phi(\mathbf{x}^1), \phi(\mathbf{x}^2), \ldots, \phi(\mathbf{x}^k)) \sim \mathcal{D}$
    Sample timestep $t \sim \mathcal{U}(1, T)$, and noise $\boldsymbol{\epsilon}^i \sim \mathcal{N}(0, I)$
    Compute noisy image $\mathbf{x}_t^i = \alpha_t \mathbf{x}^i + \sigma_t \boldsymbol{\epsilon}^i$
    Compute model scores $\mathbf{s}_i \triangleq \mathbf{s}(\mathbf{x}^i, \boldsymbol{c}, t, \boldsymbol{\theta}) = \|\boldsymbol{\epsilon}^i - \boldsymbol{\epsilon}_\theta(\mathbf{x}_t^i, \boldsymbol{c})\|_2^2 - \|\boldsymbol{\epsilon}^i - \boldsymbol{\epsilon}_{\text{ref}}(\mathbf{x}_t^i, \boldsymbol{c})\|_2^2$
    Determine ranking $\tau$ by sorting images based on $\phi(\mathbf{x}^i)$ in descending order
    **for** each pair $(i, j)$ with $i > j$ in $\tau$ **do**
        Compute pairwise gains: $G_{ij} = 2^{\phi(\mathbf{x}^i)} - 2^{\phi(\mathbf{x}^j)}$
        Compute discount factors: $D(\tau(i)) = \log(1 + \tau(i))$ and $D(\tau(j)) = \log(1 + \tau(j))$
        Compute pairwise DCG weights: $\Delta_{ij} = |G_{ij}| \cdot \left| \frac{1}{D(\tau(i))} - \frac{1}{D(\tau(j))} \right|$
        Compute pairwise loss: $\mathcal{L}_{ij} = \Delta_{i,j} \log \sigma \left( -\beta \left( \mathbf{s}(\mathbf{x}^i, \boldsymbol{c}, t, \boldsymbol{\theta}) - \mathbf{s}(\mathbf{x}^j, \boldsymbol{c}, t, \boldsymbol{\theta}) \right) \right)$
    Sum pairwise losses: $\mathcal{L}_{\text{RankDPO}} = -\sum_{i>j} \mathcal{L}_{ij}$
    Compute gradients $\text{grad}_{\text{iter}} = \nabla_\theta \mathcal{L}_{\text{RankDPO}}$
    Update model parameters: $\theta = \theta - \eta \cdot \text{grad}_{\text{iter}}$
Final $\theta^{\text{RankDPO}} = \theta$
**return** Fine-tuned model $\theta^{\text{RankDPO}}$

---

**Algorithm 3** Generate `Syn-Pic` and Train `RankDPO`

---

**Input:** $N$ prompts $(\mathcal{P} = \{\boldsymbol{c}_i\}_{i=1}^N)$, $k$ T2I Models $(\{\theta_i\}_{i=1}^k)$, $n$ Reward Models $(\{\mathcal{R}_\psi^i\}_{i=1}^n)$
**Input:** Initial model $\theta_{\text{init}}$, Reference model $\theta_{\text{ref}}$, Pre-defined signal-noise schedule $\{\alpha_t, \sigma_t\}_{t=1}^T$
**Hyper-parameters:** # Optimization Steps $(m)$, Learning Rate $(\eta)$, Divergence Control $\beta$
**Output:** Fine-tuned model $\theta^{\text{RankDPO}}$

// Generate Synthetically Labeled Ranked Preference dataset $\mathcal{D}$ using Algorithm 1
$\mathcal{D} = \text{DataGen}(\mathcal{P}, \{\theta_i\}_{i=1}^k, \{\mathcal{R}_\psi^i\}_{i=1}^n)$

// Train $\theta$ using Algorithm 2
$\theta^{\text{RankDPO}} = \text{RankDPO}(\mathcal{D}, \theta_{\text{init}}, \theta_{\text{ref}}, \{\alpha_t, \sigma_t\}_{t=1}^T, m, \eta, \beta)$
**return** Fine-tuned model $\theta^{\text{RankDPO}}$

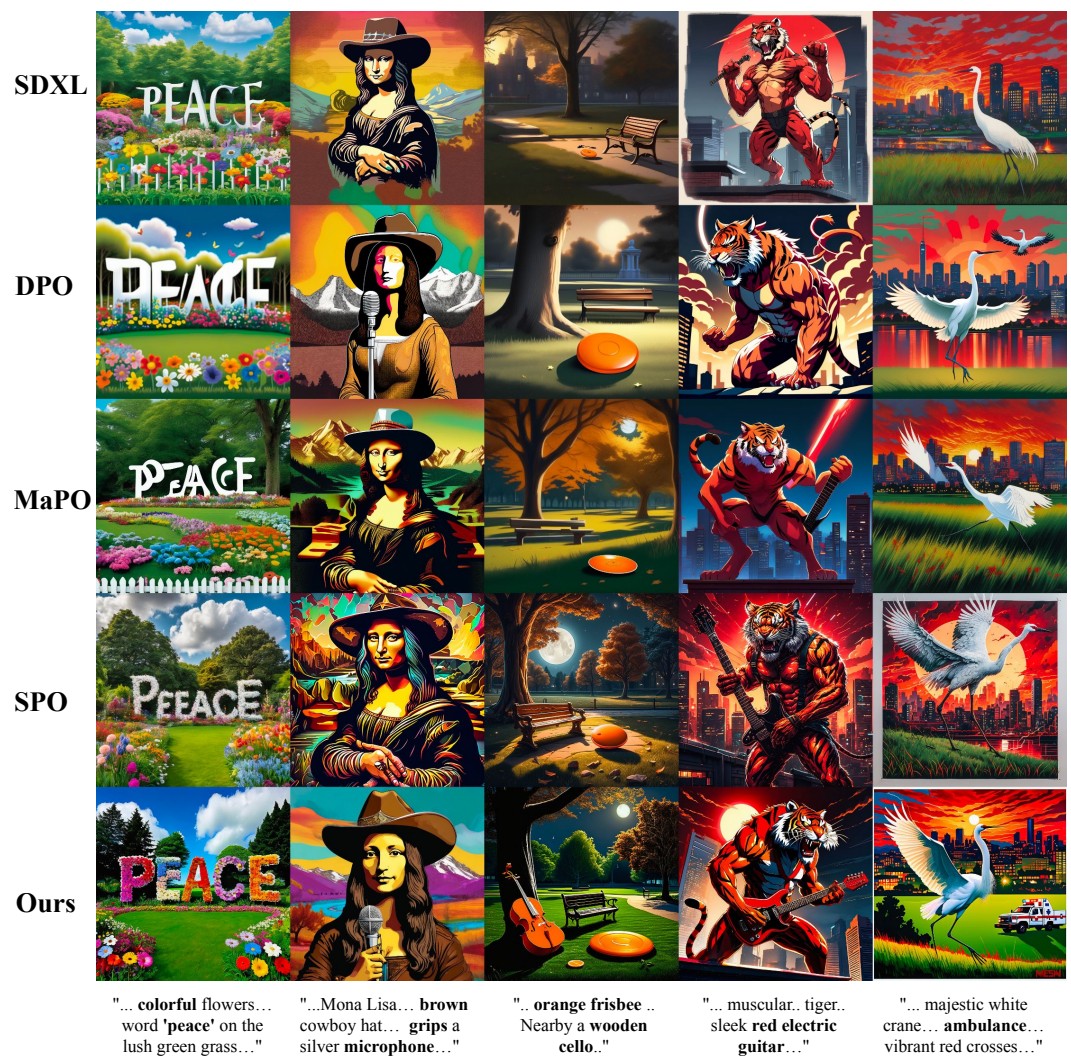

**SDXL**

**DPO**

**MaPO**

**SPO**

**Ours**

"... **colorful** flowers… word **'peace'** on the lush green grass…"

"...Mona Lisa… **brown** cowboy hat… **grips** a silver **microphone**…"

".. **orange frisbee** .. Nearby a **wooden cello**.."

"... muscular.. tiger.. sleek **red electric guitar**…"

"... majestic white crane… **ambulance**… vibrant red crosses…"

Figure 5: Comparison among different preference optimization methods and `RankDPO` for SDXL. The results illustrate that we generate images with better prompt alignment and visual quality.

## A.9 COMPLETE PROMPTS FOR FIGURES

Fig. 1 SDXL

- a vibrant garden filled with an array of colorful flowers meticulously arranged to spell out the word 'peace' on the lush green grass. The garden is enclosed by a white picket fence and surrounded by tall trees that sway gently in the breeze. Above, against the backdrop of a blue sky, whimsical clouds have been shaped to form the word 'tensions', contrasting with the tranquil scene below.

- a striking propaganda poster featuring a cat with a sly expression, dressed in an elaborate costume reminiscent of French Emperor Napoleon Bonaparte. The feline figure is holding a large, yellow wedge of cheese as if it were a precious treasure. The background of the poster is a bold red, with ornate golden details that give it an air of regal authority.

- A deserted park scene illuminated by a soft moonlight where an orange frisbee lies on the grass, slightly tilted to one side. Nearby, a wooden cello and its bow rest in solitude against a weathered park bench, their elegant forms casting long shadows on the pavement. The

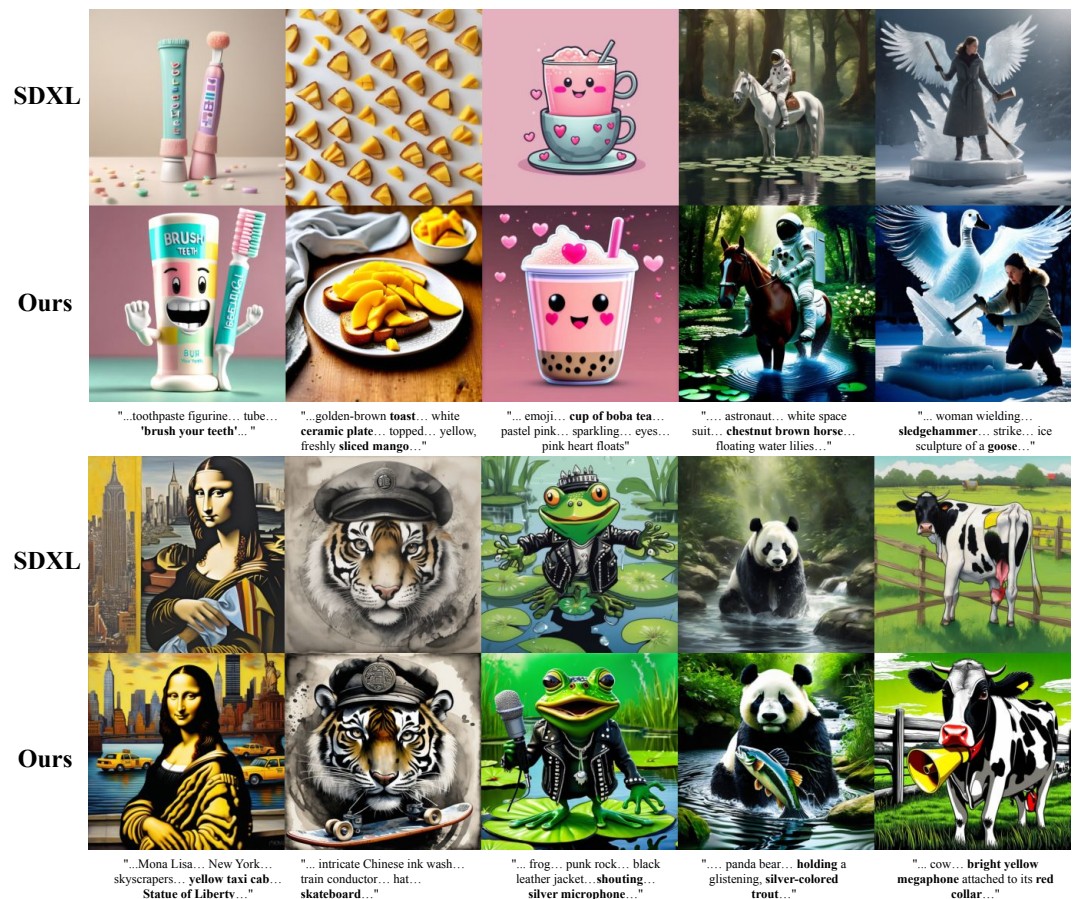

Figure 6: Qualitative comparison between SDXL, before and after our preference-tuning. The results show that our method generates images with better prompt alignment and aesthetic quality.

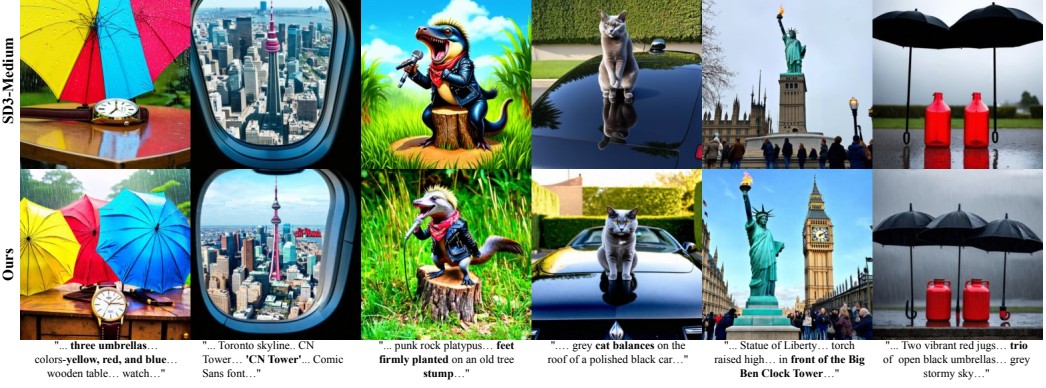

Figure 7: Qualitative comparison between SD3-Medium, before and after our preference-tuning. The results show that our method generates images with better prompt alignment and aesthetic quality.

surrounding trees sway gently in the breeze, indifferent to the forgotten items left in the wake of an earlier emergency rehearsal.

- An anime-style illustration depicts a muscular, metallic tiger with sharp, angular features, standing on a rooftop. The tiger is in a dynamic pose, gripping a sleek, red electric guitar, and its mouth is open wide as if caught in the midst of a powerful roar or song. Above the

tiger, a bright spotlight casts a dramatic beam of light, illuminating the scene and creating stark shadows on the surrounding rooftop features.

- A majestic white crane with outstretched wings captured in the act of taking flight from a patch of green grass. In the foreground, an ambulance emblazoned with vibrant red crosses races past, its siren lights ablaze with urgency against the evening sky. The cityscape beyond is silhouetted by the fading hues of dusk, with the outlines of buildings casting long shadows as the day comes to a close.

Fig. 1 SD3

- A beautifully aged antique book is positioned carefully for a studio close-up, revealing a rich, dark brown leather cover. The words "Knowledge is Power" are prominently featured in the center with thick, flowing brushstrokes, gleaming in opulent gold paint. Tiny flecks of the gold leaf can be seen scattered around the ornately scripted letters, showcasing the craftsmanship that went into its creation. The book is set against a plain, uncluttered background that focuses all attention on the intricate details of the cover's design.

- A pristine white bird with a long neck and elegant feathers stands in the foreground, with a towering dinosaur sculpture positioned behind it among a grove of trees. The dinosaur, a deep green in color with textured skin, contrasts sharply with the smooth plumage of the bird. The trees cast dappled shadows on the scene, highlighting the intricate details of both the bird and the prehistoric figure.

- A striking portrait photograph showcasing a fluffy, cream-colored hamster adorned with a vibrant orange beanie and oversized black sunglasses. The hamster is gripping a small white sign with bold black letters that proclaim "Let's PAINT!" The background is a simple, blurred shade of grey, ensuring the hamster remains the focal point of the image.

- A whimsical scene unfolds in a lecture hall where a donkey, adorned in a vibrant clown costume complete with a ruffled collar and a pointed hat, stands confidently at the podium. The donkey is captured in a high-resolution photo, addressing an audience of attentive students seated in rows of wooden desks. Behind the donkey, a large blackboard is filled with complex mathematical equations, hinting at the serious nature of the lecture juxtaposed with the humorous attire of the lecturer.

- A spacious living room features an unlit fireplace with a sleek, flat-screen television mounted above it. The television screen displays a heartwarming scene of a lion embracing a giraffe in a cartoon animation. The mantle of the fireplace is adorned with decorative items, including a small clock and a couple of framed photographs.

- An ornate representation of the Taj Mahal intricately positioned at the center of a gold leaf mandala, which showcases an array of symmetrical patterns and delicate filigree. Surrounding the central image, the mandala's design features accents of vibrant blues and reds alongside the gold. Below this striking visual, the words "Place of Honor" are inscribed in an elegant, bold script, centered meticulously at the bottom of the composition.

Fig. 4

- A plump wombat, adorned in a crisp white panama hat and a vibrant floral Hawaiian shirt, lounges comfortably in a bright yellow beach chair. In its paws, it delicately holds a martini glass, the drink precariously balanced atop the keys of an open laptop resting on its lap. Behind the relaxed marsupial, the silhouettes of palm trees sway gently, their forms blurred into the tropical backdrop.

- a whimsical scene featuring a bright orange fruit donning a miniature brown cowboy hat with intricate stitching. The orange sits atop a wooden table, its textured peel contrasting with the smooth surface beneath. To the side of the orange, there's a small cactus in a terracotta pot, completing the playful western theme.

- A creative studio photograph featuring tactile text spelling 'hello' with vibrant, multicolored fur that stands out boldly against a pure white background. This playful image is showcased within a unique frame made of equally fluffy material, mimicking the texture of the centerpiece. The whimsical arrangement is perfectly centered, lending a friendly and inviting vibe to the viewer.

- An intricately detailed oil painting depicts a raccoon dressed in a black suit with a crisp white shirt and a red bow tie. The raccoon stands upright, donning a black top hat and gripping a wooden cane with a silver handle in one paw, while the other paw clutches a dark garbage bag. The background of the painting features soft, brush-stroked trees and mountains, reminiscent of traditional Chinese landscapes, with a delicate mist enveloping the scene.

- A vibrant yellow rabbit, its fur almost glowing with cheerfulness, bounds energetically across a sprawling meadow dotted with a constellation of wildflowers. The creature's size-able, red-framed glasses slip comically to the tip of its nose with each jubilant leap. As the first rays of sunlight cascade over the horizon, they illuminate the dew-draped blades of grass, casting the rabbit's exuberant shadow against the fresh green canvas.

- A whimsical scene unfolds in a lecture hall where a donkey, adorned in a vibrant clown costume complete with a ruffled collar and a pointed hat, stands confidently at the podium. The donkey is captured in a high-resolution photo, addressing an audience of attentive students seated in rows of wooden desks. Behind the donkey, a large blackboard is filled with complex mathematical equations, hinting at the serious nature of the lecture juxtaposed with the humorous attire of the lecturer.

Fig. 5

- a vibrant garden filled with an array of colorful flowers meticulously arranged to spell out the word 'peace' on the lush green grass. The garden is enclosed by a white picket fence and surrounded by tall trees that sway gently in the breeze. Above, against the backdrop of a blue sky, whimsical clouds have been shaped to form the word 'tensions', contrasting with the tranquil scene below.

- a reimagined version of the Mona Lisa, where the iconic figure is depicted with a brown cowboy hat tilted rakishly atop her head. In her hand, she grips a silver microphone, her mouth open as if caught mid-scream of a punk rock anthem. The background, once a serene landscape, is now a vibrant splash of colors that seem to echo the intensity of her performance.

- A deserted park scene illuminated by a soft moonlight where an orange frisbee lies on the grass, slightly tilted to one side. Nearby, a wooden cello and its bow rest in solitude against a weathered park bench, their elegant forms casting long shadows on the pavement. The surrounding trees sway gently in the breeze, indifferent to the forgotten items left in the wake of an earlier emergency rehearsal.

- An anime-style illustration depicts a muscular, metallic tiger with sharp, angular features, standing on a rooftop. The tiger is in a dynamic pose, gripping a sleek, red electric guitar, and its mouth is open wide as if caught in the midst of a powerful roar or song. Above the tiger, a bright spotlight casts a dramatic beam of light, illuminating the scene and creating stark shadows on the surrounding rooftop features.

- A majestic white crane with outstretched wings captured in the act of taking flight from a patch of green grass. In the foreground, an ambulance emblazoned with vibrant red crosses races past, its siren lights ablaze with urgency against the evening sky. The cityscape beyond is silhouetted by the fading hues of dusk, with the outlines of buildings casting long shadows as the day comes to a close.

Fig. 6

- A digitally rendered image of a whimsical toothpaste tube figurine that boasts a candy pastel color palette. The figurine is set against a soft, neutral background, enhancing its playful charm. On the body of the toothpaste tube, bold letters spell out the reminder 'brush your teeth,' inviting a sense of dental care responsibility. The tube cap is carefully designed to exhibit a realistic, shiny texture, creating a striking contrast with the matte finish of the tube itself.

- A piece of golden-brown toast resting on a white ceramic plate, topped with bright yellow, freshly sliced mango. The mango slices are arranged in a fan-like pattern, and the plate sits on a light wooden table with a few crumbs scattered around. The texture of the toast contrasts with the soft, juicy mango pieces, creating an appetizing snack.

- An intricately designed digital emoji showcasing a whimsical cup of boba tea, its surface a glistening shade of pastel pink. The cup is adorned with a pair of sparkling, heart-shaped eyes and a curved, endearing smile, exuding an aura of being lovestruck. Above the cup, a playful animation of tiny pink hearts floats, enhancing the emoji's charming appeal.

- An intricately designed digital emoji showcasing a whimsical cup of boba tea, its surface a glistening shade of pastel pink. The cup is adorned with a pair of sparkling, heart-shaped eyes and a curved, endearing smile, exuding an aura of being lovestruck. Above the cup, a playful animation of tiny pink hearts floats, enhancing the emoji's charming appeal.

- A surreal image capturing an astronaut in a white space suit, mounted on a chestnut brown horse amidst the dense greenery of a forest. The horse stands at the edge of a tranquil river, its surface adorned with floating water lilies. Sunlight filters through the canopy, casting dappled shadows on the scene.

- a focused woman wielding a heavy sledgehammer, poised to strike an intricately carved ice sculpture of a goose. The sculpture glistens in the light, showcasing its detailed wings and feathers, standing on a pedestal of snow. Around her, shards of ice are scattered across the ground, evidence of her previous strikes.

- A detailed painting that features the iconic Mona Lisa, with her enigmatic smile, set against a bustling backdrop of New York City. The cityscape includes towering skyscrapers, a yellow taxi cab, and the faint outline of the Statue of Liberty in the distance. The painting merges the classic with the contemporary, as the Mona Lisa is depicted in her traditional attire, while the city behind her pulses with modern life.

- An intricate Chinese ink and wash painting that depicts a majestic tiger, its fur rendered in delicate brush strokes, wearing a traditional train conductor's hat atop its head. The tiger's piercing eyes gaze forward as it firmly grasps a skateboard, which features a prominent yin-yang symbol in its design, symbolizing balance. The background of the painting is a subtle wash of grays, suggesting a misty and timeless landscape.

- An animated frog with a rebellious punk rock style, clad in a black leather jacket adorned with shiny metal studs, is energetically shouting into a silver microphone. The frog's vibrant green skin contrasts with the dark jacket, and it stands confidently on a large green lily pad floating on a pond's surface. Around the lily pad, the water is calm, and other pads are scattered nearby, some with blooming pink flowers.

- A sizable panda bear is situated in the center of a bubbling stream, its black and white fur contrasting with the lush greenery that lines the water's edge. In its paws, the bear is holding a glistening, silver-colored trout. The water flows around the bear's legs, creating ripples that reflect the sunlight.

- In a grassy field stands a cow, its fur a patchwork of black and white, with a bright yellow megaphone attached to its red collar. The grass around its hooves is a lush green, and in the background, a wooden fence can be seen, stretching into the distance. The cow's expression is one of mild curiosity as it gazes off into the horizon, the megaphone positioned as if ready to amplify the cow's next "moo".

Fig 7

- On a rainy day, three umbrellas with bright and varied colors—yellow, red, and blue—are opened wide and positioned upright on a worn, wooden table. Their fabric canopies are dotted with fresh raindrops, capturing the soft, diffused light of a hazy morning. Beside these umbrellas lies a classic round watch with a leather strap and a polished face that reflects the muted light. The watch and umbrellas share the table's space, hinting at a paused moment in a day that has just begun.

- An aerial view of Toronto's skyline dominated by the iconic CN Tower standing tall amongst the surrounding buildings. The image is taken from the window of an airplane, providing a clear, bird's-eye perspective of the urban landscape. Across the image, the words "The CN Tower" are prominently displayed in the playful Comic Sans font. The cluster of city structures is neatly bisected by the glistening blue ribbon of a river.

- A vibrant scene featuring a punk rock platypus, its webbed feet firmly planted on an old tree stump. The creature is clad in a black leather jacket, embellished with shiny metal

studs, and it's passionately shouting into a silver microphone. Around its neck hangs a bright red bandana, and the stump is situated in a small clearing surrounded by tall, green grass.

- A sleek gray cat balances on the roof of a polished black car. The car is situated in a driveway, flanked by neatly trimmed hedges on either side. Sunlight reflects off the car's surface, highlighting the cat's poised stance as it surveys its surroundings.

- The iconic Statue of Liberty, with its verdant green patina, stands imposingly with a torch raised high in front of the Big Ben Clock Tower, whose clock face is clearly visible behind it. The Big Ben's golden clock hands contrast against its aged stone façade. In the surrounding area, tourists are seen marveling at this unexpected juxtaposition of two renowned monuments from different countries.

- Two vibrant red jugs are carefully positioned below a trio of open black umbrellas, which stand stark against the backdrop of a grey, stormy sky. The jugs rest on the wet, glistening concrete, while the umbrellas, with their smooth, nylon fabric catching the breeze, provide a sharp contrast in both color and texture. Each umbrella casts a protective shadow over the jugs, seemingly safeguarding them from the impending rain.

