# OpenReview forum: "Scalable Ranked Preference Optimization for Text-to-Image Generation"
_ICLR.cc/2025/Conference — Submitted to ICLR 2025_

### Official Review · Reviewer_9PJP · 2024-10-29

**Soundness:** 3
**Presentation:** 3
**Contribution:** 2
**Rating:** 5
**Confidence:** 4

**Summary:**

This paper introduces a powerful and cost-effective approach for the preference optimization of text-to-image models. Specifically, this work demonstrates how synthetically generating a preference optimization dataset can enable the collection of superior signals (e.g., rankings vs. pairwise preferences and ensembling preferences across models). Moreover, this paper also introduces a simple method to leverage the stronger signals, leading to state-of-the-art results on various benchmarks for prompt following and visual quality for both the diffusion and rectified flow models.

**Strengths:**

Inspired by the LLMs community, aligning T2I models with human feedback has become an important and practical topic to enhance existing T2I models. However, there are two major efforts in this area: 1) collecting large amounts of user preferences images for training and 2) fine-tuning with T2I models with reward functions. The former direction shows promising results when utilizing Direct Preference Optimization (DPO), but with expensive data collection cost, e.g., Pick-a-Picv2 costs nearly $50K. The later direction fine-tunes the T2I models by maximizing the reward functions with the generated images, but this process is computationally expensive and suffers from “reward hacking”, where this optimization process increases the reward scores without improving the quality of the generated images.

This paper address the above challenges and propose a scalable and cost-effective solution for aligning T2I models, which includes the following two contributions:
- Synthetically Labeled Preference Dataset (Syn-Pic).
- Ranking-based Preference Optimization (RankDPO).

The experimental results seem to consistently show the advantages of this approach, indicating that it is an effective solution.

**Weaknesses:**

- **Lack of comparison of training costs.**  Cost-efficiency is the main claimed contribution of this work, but the paper only briefly mentions the monetary comparison (\\$200 v.s. \\$50K for Pick-a-Picv2). I believe that the gain of this work lies in the newly proposed loss function $\mathcal{L}_{\text{RankDPO}}$, so its training cost comparison, e.g., training time, gpus, flopts, and so on, should be included in main text and Tab.8.

- **Clarification of the ``*Reference model*'' definition**. The ``*reference model*'' defined in the method is not fully explained in lines 204 and 285, resulting in a less than intuitive understanding of the method.

- **Analysis of the effectiveness and limitations of Ground Truth.** I understand that Eq.5 is the core of the proposed RankDPO, which means that the model is supervised based on the reference model or the so-called ground truth (GT) preference, so I really want to know how this GT preference is obtained (Section 3.2 does not seem to mention)? Is there a subjective bias for GT that causes the performance of the model to have an upper limit (e.g., only select few intended t2I models)? Does the scale k of the adopted t2I models have an impact on the ground truth?

- **Lack of *scalable* explanations and quantitative or qualitative comparisons.** In line 92, it is claimed that this work is scalable, but there is a lack of theoretical explanation, qualitative analysis and quantitative comparison of scalability in the experiments and methods. In other words, are all previous methods not scalable? How to measure the scalability of this work? Can this work be adapted to the preference annotation of other tasks such as video and 3D? What is the connotation of this scalability? Please give a detailed explanation and clear definition.

**Questions:**

See weaknesses.

---

> ### Author Response · Authors · 2024-11-20
>
> We thank the reviewer for taking the time to provide a detailed review of our work. We address the concerns raised by the reviewer below:
>
> **Clarification of Reference Model:** We thank the reviewer for pointing out the lack of clarity. We have updated the paper (Sec. 3.1) to describe this process in more detail. Briefly, in Reinforcement Learning from Human Feedback (RLHF), we aim to maximize a reward function $r(.)$, which in our case are models that rate how good a generated image is for a given text prompt. However, naively optimizing the reward can lead to collapse. Therefore, RLHF methods enforce a KL-regularization to the *original* base model from which we start the optimization process. This original model checkpoint, i.e the starting point, is referred to as the reference model, and the weight of the KL-regularization term controls how much we change from the original/reference model.
>
> **Effectiveness and Limitations of Ground Truth:** In Eq. 5 (Eq. 6 in the updated paper), we have the model ($\epsilon_\theta$) that is being trained, the original reference model ($\epsilon_\text{ref}$), and the winning and losing sample. In our case, the winning and losing samples are decided based on off-the-shelf reward models. In earlier works, these samples are collected using human-labeling, i.e., user-collected data. Naturally, user-collected data can be noisy, and several off-the-shelf reward models have shown a better correlation to human judgment than a *single* user. Further, we decide the winning and losing sample by aggregating scores/votes from five reward models, making the "ground truth" fairly robust. The role of the reference (original) model in Eq. 5 is to help measure how much the denoising is improved for the model when compared to the reference/original model that we start from. The overall goal of DPO in diffusion models is to improve the denoising for the preferred (winning) samples while reducing the denoising for the less preferred (losing) samples when compared to the reference/original model.
>
> **Explanations behind Scalability:** Our primary motivation behind this work was to be able to construct an effective preference dataset for enhancing text-to-image models. The biggest bottleneck in this process is collecting, verifying, and performing quality control of human annotations. We show that purely synthetic preferences from off-the-shelf reward models provide an effective alternate solution to the costly human-labeled datasets. This allows us to arbitrarily scale up the size of preference datasets for text-to-image models. We also show that our dataset Syn-Pic allows for far superior results than the existing Pick-a-Picv2 dataset even though it has 1/4th the images, showcasing its effectiveness.  In this work, we use the same prompts set used in Pick-a-Picv2 dataset for a fair comparison. However, we  can create a much larger preference dataset by increasing the size of the prompts set, with significantly lower annotation cost than human labeling (cheaper by *2-3 orders of magnitude*). Therefore, we wish to see future work building upon our work and increasing the size and quality of future preference datasets.
>
> **Extensions to Videos/3D:** The reviewer also makes an important point about extending this framework for text-to-{video, 3D} models. We would like to point out that recent reward models (e.g., VQAScore) have shown promise for aligning with human preferences for videos and 3D modalities. Therefore, our framework would be a promising option for curating preference datasets for enhancing video/3D models.
>
> **Comparison of training costs:** We already discuss training cost for RanDPO in Sec. 4.2 (Discussion of Computation Cost) and Tab. 8 in Appendix. In terms of the training cost, the only overhead RankDPO has over the standard DPO objective is computing the DCG weights. In our case, this amounts to an argsort operation over four entries, which does not have a meaningful overhead. However, due to the improvements of our Syn-Pic dataset both in image and labeling quality, we require fewer iterations until convergence (since the dataset has 4x fewer images), while providing superior results on all downstream evaluations. We provide this comparison in Tab. 8 of the paper.
>
>
> | **Method**   | **Training Images** | **A100 GPU Days** | **DPG-Bench Score** |
> |--------------|---------------------|--------------------|---------------------|
> | DPO      	| 1.0M           	| 30             	| 76.74          	|
> | MaPO     	| 1.0M           	| 25             	| 74.53          	|
> | SPO      	| -              	| 5              	| 74.73          	|
> | Ours     	| 0.24M          	| 6              	| 79.26          	|

---

> > ### Author Response · Authors · 2024-11-25
> >
> > Dear Reviewer 9PJP,
> >
> > We would like to thank you again for your valuable feedback on our paper. We have included additional experiments and clarifications to address your concerns. We have added clarifications regarding the reference model and ground truth in DPO formulation, scalability of our approach, and extensions to other modalities. We have also included the training cost comparison of our work to other baselines. We have also included other experiments to show the benefits of our approach (see reply in main thread https://openreview.net/forum?id=Y6KUBkUimC&noteId=t0TJetYvG9 ).
> >
> > As the period for the Author-Reviewer discussion is closing very soon, we would like to use this opportunity to kindly ask if our responses sufficiently clarify your concerns. We sincerely appreciate your time and consideration.
> >
> > Best Regards, Authors

---

> > > ### Author Response · Authors · 2024-12-02
> > >
> > > Dear Reviewer 9PJP,
> > >
> > > We would like to thank you again for your valuable feedback on our paper.
> > >
> > > As the period for the Author-Reviewer discussion is closing very soon, we would like to use this opportunity to kindly ask if our responses sufficiently clarify your concerns. We sincerely appreciate your time and consideration.
> > >
> > > Best Regards, Authors

---

### Official Review · Reviewer_Pz5f · 2024-10-30

**Soundness:** 3
**Presentation:** 3
**Contribution:** 2
**Rating:** 5
**Confidence:** 4

**Summary:**

This paper proposes a scalable approach to fine-tune text-to-image generation (T2I) models using DPO without requiring human annotation. The authors introduce a synthetic preference dataset collection method, where preferences are generated using off-the-shelf reward models, eliminating the need for human annotation. Additionally, this approach leverages model predictions across multiple models, enabling dense and ranked preferences rather than just parse pairwise ones. The authors also propose RankDPO, a framework that incorporates ranking feedback to enhance the DPO process. Their method, applied to SDXL and SD3-Medium models, demonstrates improved prompt alignment and visual quality on various benchmarks. This study contributes a cost-effective, efficient pipeline to advance T2I model alignment.

**Strengths:**

+ Scalable and cost-effective dataset collection with synthetic preferences.

+ The proposed RankDPO framework effectively leverages ranked feedback to improve text-image alignment

+ Experiments on benchmark datasets could verify the effectiveness of the proposed method.

**Weaknesses:**

- Lack of novelty: Existing work (Wallace et al. 2024, Liu et al. 2024c) have proposed diffusion-based DPO and List-wise ranking-based DPO. It seems that the proposed method combines them to tune diffusion models.

- More baseline methods, especially those used to generate images in the process of the construction of Syn-Pic, should be considered and compared in Tab 1, Tab 2, and Tab 3.

- The performance improvement gained by RankDPO over DPO (5 Rewards) in Tab 4 seems marginal. The effectiveness of the list-wise or rank-wise DPO method may be not strongly verified.

Minors:
It seems to be “Tab. 5” instead of “Tab. 4” in line 425.

**Questions:**

- The improvement in the non-spatial category of T2I-CompBench, as shown in Tab 2, seems not remarkable compared with others. Why is that and how to explain it?

- How does the proposed method influence the quality of generated images (e.g., FID), although the alignment has been improved as shown in Tab 1-3?

- Is it possible that the proposed method just “distills” the useful knowledge from the models used to generated images in the process of the construction of Syn-Pic, especially those strong base models?

---

> ### Author Response · Authors · 2024-11-20
>
> We thank the reviewer for taking the time to provide thoughtful feedback on our paper. We address the concerns raised by the reviewer below:
>
> **Novelty:** We describe the novel aspects of our work below.
>
> **Synthetically Labeled Preference Dataset (Syn-Pic)**
> As described in the introduction, the biggest bottleneck of existing DPO works (e.g., DiffusionDPO) is the lack of a high-quality preference dataset. Human annotation of preference datasets is both costly and cumbersome. Additionally, these datasets quickly become outdated due to the rapid advancements in T2I model capabilities. For instance, the Pick-A-Picv2 dataset, which was manually labelled, primarily contains 512-resolution images produced by variants of the SD1.5 model. This has already been surpassed by newer T2I models such as SDXL and SD3.
>
> In this work, we propose replacing human labeling with reward model scores to quickly iterate on new preference datasets. This approach provides several benefits:
> - **No human labeling required:** Avoids noisy labels and reduces costs significantly.
> - **Stronger Preference Signal:** Allows for aggregating preferences from multiple models and generating ranked preference lists for the same prompt.
> - **Scalability:** Enables the use of a much larger prompt set, as annotation costs are drastically reduced.
>
> **Ranking-based Preference Optimization (RankDPO):**
> Earlier DPO works for T2I models only had access to pairwise comparisons. However, our Syn-Pic dataset provides ranked preference lists of images for each prompt. This enables us to propose an updated direct preference optimization method that incorporates the ranking objective into its formulation.
>
> **Extensive Experiments to Show Benefits of Syn-Pic and RankDPO:**
> - **Syn-Pic + RankDPO improves T2I models:** Syn-Pic combined with RankDPO significantly improves the performance of models such as SDXL and SD3 (see Tables 1, 2, 3). Even though the publicly available SD3 model has already undergone DPO training with a larger dataset, our proposed method further enhances its performance.
> - **Individual contributions yield non-trivial gains:** Both components individually lead to notable improvements (see Table 4).
>   - DPO(Pick-a-Picv2) vs DPO(5 rewards): Demonstrates the superiority of the Syn-Pic dataset over the Pick-a-Picv2 dataset.
>   - DPO(5 rewards) vs RankDPO: Shows the improvements of RankDPO over DiffusionDPO. This gap widens when using the GenEval benchmark (see Table 6 in the updated paper).
>
> **Baseline Methods:** We thank the reviewer for pointing this out. For Syn-Pic, we generate images from SDXL, SD3-Medium, Stable Cascade, and PixArt-Sigma. We have now updated Tables 1-3 to include results from all four models. We see the same trends as before : (a) SD3-Medium is the strongest model, and (b) consistent improvements are provided to SDXL and SD3-Medium by our method.
>
> **Quality/FID evaluation:** We thank the reviewer for bringing up this aspect of evaluation. We predominantly focused on evaluating visual quality through the Q-Align aesthetic score model and user studies. Fig. 3 shows that users consistently prefer our model over SDXL/DPO-SDXL.
> However, we now also perform an analysis of the FID scores using the MJHQ-30K benchmark, by comparing the FID against a set of 30K high-quality images from Midjourney. Existing SDXL+Refiner model achieves an FID of $9.55$, while our model achieves an FID of $7.99$. We also provide the full table in Tab. 10 of the paper.
>
> | **Category**   	| **Animals** | **Art**   | **Fashion** | **Food**  | **Indoor** | **Landscape** | **Logo**   | **People** | **Plants**  | **Vehicles** | **Overall** |
> |---------------------|-------------|-----------|-------------|-----------|------------|---------------|------------|------------|-------------|--------------|-------------|
> | **SDXL-Refiner**	| 28.93   	| 31.05 	| 28.90   	| 30.09 	| 28.83  	| 30.78     	| 36.67  	| 35.56  	| **28.42**   | 24.45    	| 9.55    	|
> | **RankDPO SDXL**	| **24.37**   | **27.22** | **22.91**   | **25.02** | **26.01**  | **28.61** 	| **29.74**  | **27.54**  | 30.27   	| **21.83**	| **7.99**	|
>
> **Benefits from the ranking objective:** We agree that the biggest source of performance gains is from preference-tuning on our Syn-Pic dataset, a key contribution of the paper. However, we would point out that the increase in performance from our RankDPO is a significant improvement. We further evaluate this on GenEval where we continue to see consistent improvements for SDXL and SD3-Medium (Tab. 6 of the updated paper)

---

> > ### Author Response · Authors · 2024-11-20
> >
> > | **Model**    	| **Mean ↑** | **Single ↑** | **Two ↑** | **Counting ↑** | **Colors ↑** | **Position ↑** | **Color Attribution ↑** |
> > |-------------------|------------|---------------|------------|-----------------|---------------|-----------------|--------------------------|
> > | SDXL         	| 0.55   	| 0.98      	| 0.74   	| 0.39        	| 0.85      	| 0.15        	| 0.23                 	|
> > | DPO	| 0.59   	| 0.99      	| 0.84   	| 0.49        	| 0.87      	| 0.13        	| 0.24                 	|
> > | **RankDPO**   	| **0.61**   | 1.00  	| 0.86   | 0.46    	| 0.90  	| 0.14    	| 0.29             	|
> > |-------------------|------------|---------------|------------|-----------------|---------------|-----------------|--------------------------|
> > | SD3-Medium   	| 0.70   	| 1.00      	| 0.87   	| 0.63        	| 0.84      	| 0.28        	| 0.58                 	|
> > | DPO	| 0.72   	| 1.00      	| 0.90   	| 0.63        	| 0.87      	| 0.32        	| 0.58                 	|
> > | **RankDPO**   	| **0.74**   | 1.00  	| 0.91   | 0.72    	| 0.87  	| 0.31    	| 0.66             	|
> >
> >
> > **T2I-Compbench Non-Spatial:** The non-spatial category for T2I-Compbench is computed using CLIPScore, unlike attribute binding, which computes the percentage of correct attributes with a VQA model. In general, CLIP scores tend to be in the same range without too much of a difference. For instance, going from SD1.4 to SD3-Medium sees a drastic improvement in all evaluations but only increases the CLIP score from 30.79 to 31.25.
> >
> > **Are we just distilling superior images?** In Tab. 4, we have an (only SDXL) entry where we repeated the whole procedure purely with SDXL images and then applied RankDPO to SDXL on this dataset. We still see consistent improvements, which highlights that synthetic preference labeling and RankDPO can be used to improve the quality of output of a model even without a superior model to provide training data.
> >
> > | **Model Name**         	| **DSG Score** | **VQA Score** | **Q-Align Score** |
> > |-----------------------------|---------------|---------------|-------------------|
> > | SDXL                   	| 74.65     	| 84.33     	| 0.72          	|
> > | **RankDPO (Only SDXL)** 	| 78.40 	| 86.76 	| 0.74      	|
> > | **RankDPO**            	| 79.26 	| 87.52 	| 0.81      	|

---

> > > ### Author Response · Authors · 2024-11-25
> > >
> > > Dear Reviewer Pz5f,
> > >
> > > We would like to thank you again for your valuable feedback on our paper. We have included additional experiments and clarifications to address your concerns. The experiments include:
> > > - (a) PixArt-Sigma and Stable Cascade baselines in Tab.1/2/3,
> > > - (b) FID evaluation on MJHQ-30K dataset,
> > > - (c) Ablations on DPO vs RankDPO on Syn-Pic dataset, and
> > > - (d) RankDPO on just SDXL synthetic dataset.
> > >
> > > We have also elaborated a bit more on novel aspects of our submission.
> > >
> > > As the period for the Author-Reviewer discussion is closing very soon, we would like to use this opportunity to kindly ask if our responses sufficiently clarify your concerns. We sincerely appreciate your time and consideration.
> > >
> > > Best Regards, Authors

---

> > > > ### Author Response · Authors · 2024-12-02
> > > >
> > > > Dear Reviewer Pz5f,
> > > >
> > > > We would like to thank you again for your valuable feedback on our paper. We have included additional experiments and clarifications to address your concerns.
> > > >
> > > > As the period for the Author-Reviewer discussion is closing very soon, we would like to use this opportunity to kindly ask if our responses sufficiently clarify your concerns. We sincerely appreciate your time and consideration.
> > > >
> > > > Best Regards, Authors

---

### Official Review · Reviewer_GRrX · 2024-11-02

**Soundness:** 3
**Presentation:** 3
**Contribution:** 3
**Rating:** 8
**Confidence:** 4

**Summary:**

Overall, this is a very good and solid paper. The paper addresses the challenges of applying Direct Preference Optimization (DPO) to text-to-image (T2I) models, particularly the resource-intensive process of collecting human preference datasets. The authors propose a scalable approach for creating large-scale synthetic datasets for DPO training, using a pre-trained reward function to generate preferences for paired images. This method allows for averaging predictions across multiple models and collecting ranked preferences instead of just pairwise preferences. The researchers introduce RankDPO, an enhancement to DPO-based methods that utilizes ranking feedback. Applying RankDPO on SDXL and SD3-Medium models with the synthetic "Syn-Pic" dataset improves both prompt-following and visual quality, offering a practical solution to enhance the performance and safety of T2I models.

**Strengths:**

1. The writing of this paper is clear and easy to follow. It's easy to understand the main contributions of this paper as I summarized above.
2. The contributions of this paper are solid. [Method contribution] The authors propose a new performance optimization method RankDPO, and the effectiveness is well supported by extensive experiments on SDXL, SD3-Medium models and GenEval, T2I-CompBench benchmarks. [Dataset contribution] To make the ranking-based preference optimization work, the authors collect a new dataset Synthetically Labeled Preference Dataset (Syn-Pic), which can benefit future research in text-to-image community.

**Weaknesses:**

No obvious weaknesses. Please check the questions section.

**Questions:**

1. How to evaluate the reliability of the Reward Models when building the Syn-Pic dataset? Do you conduct a human evaluation study, for example, do you calculate the correlation between the reward model ranking and human evaluation ranking?
2. There are some other text-to-image benchmarks as I know, for example, HRS-bench [*]. Can you explain the reason why not include other benchmarks in your paper?

[*] Bakr, E. M., Sun, P., Shen, X., Khan, F. F., Li, L. E., & Elhoseiny, M. (2023). Hrs-bench: Holistic, reliable and scalable benchmark for text-to-image models. In Proceedings of the IEEE/CVF International Conference on Computer Vision (pp. 20041-20053).

---

> ### Author Response · Authors · 2024-11-20
>
> We thank the reviewer for their positive evaluation and recognition of the solid contributions of our work. Below, we address the questions raised by the reviewer:
>
> **Choice of Evaluation Benchmarks:** As the reviewer indicates, there are several benchmarks for evaluating text-to-image models (e.g., GenEval, T2I-Compbench, GenAI Bench, ConceptMix, HRS-Bench, TIFA). While we would be keen on adding more evaluations to further strengthen our claims, our choice of evaluation benchmarks were driven by the following criteria:
>
> a. Having several top-performing recent models already evaluated,
>
> b. Focus on  accurately evaluating prompt-following and visual quality with a variety of evaluation strategies such as using object detectors and visual question answering models.
>
> Our choice of evaluation benchmarks (GenEval, DPG-Bench, T2I-Compbench) have been used very frequently in the recent state-of-the-art T2I models to judge their quality. It includes models such as DALL-E 3, SDXL, SD-3, PixArt, SANA, etc. Further, it also reduces our computational burden since previous works have already evaluated and scored their T2I model on these benchmarks.
>
> **Evaluation of Reward Models:** The reviewer brings up the critical question of how we evaluate reward models in text-to-image generation, especially given their growing usage in preference optimization and evaluation. Typically, reward models at the time of introduction are benchmarked on a variety of *small* evaluation sets which have human preferences and therefore can be used to measure correlation with human judgment. For instance, HPSv2.1, ImageReward, MPS, and Pickscore have been evaluated on the HPD, ImageReward and Pick-a-Pic test sets where these models show good correlation with human judgment (in some cases even outperforming individual users). Similarly, VQAScore has also been evaluated on several benchmarks where not only does it show a high quality of measuring image-text alignment, but is also overall aligned with human preferences. Based on these experiments, we decided to choose these models, and opted for an ensemble strategy where we aggregate preferences from these five models to improve the robustness of our synthetically generated preferences.

---

> > ### Comment · Reviewer_GRrX · 2024-11-24
> >
> > Thank you for your rebuttal, and it has addressed my concerns. I keep my positive rating.

---

> > > ### Author Response · Authors · 2024-11-25
> > >
> > > Dear Reviewer GRrX,
> > >
> > > Thank you so much for your time and efforts. We greatly appreciate your constructive feedback and positive rating.
> > >
> > > Best Regards, Authors

---

### Official Review · Reviewer_Rfve · 2024-11-03

**Soundness:** 2
**Presentation:** 2
**Contribution:** 3
**Rating:** 6
**Confidence:** 4

**Summary:**

This paper studies the preference optimization of T2I diffusion models, particularly SD3-Medium and SDXL. The main contribution made by this paper has two folds. 1) This paper proposed a new synthetic preference dataset containing rankings annotated by existing preference models, which cost less than similar datasets with human annotation. 2) Based on the ranking-based dataset, this paper proposes a new DPO method using the ranking information to improve the prompt following and generation quality of T2I diffusion models.

**Strengths:**

- The overall presentation of this paper is clear.
- I like the idea that extending existing human preference dataset with (win, lose) tuple to having more items, naturally leading to a rank based preference dataset.
- The experiments are conducted on both big and medium sized models: SDXL and SD3-Medium, which is adequate. The comparison experiments with other models finetuned with DPO and different preference datasets show that the proposed new dataset and method has better performance than others.

**Weaknesses:**

- Overall, I like the idea of using rank-based preference dataset and DPO, which should be able to provide more fine-grained preference guidance than (win, lose) style DPO and datasets. My only concern is that it is unclear whether the performance improvement of the proposed RankDPO is more because of the new synthetic dataset or the rank-based DPO. According to my understanding, in Table 3, DPO-SDXL, MaPO SDXL, SPO SDXL, and the proposed RankDPO SDXL are fine-tuned with different preference datasets, which mixes the impact of the new dataset and new DPO model. It would be better if the author could provide the insight about whether dataset or the rank-based DPO, is more important to the performance improvement over other preference optimization models. What if train DPO/SPO/MaPO with newly collected rank preference dataset? For example, randomly pick two images from a rank set and set the one with the lower rank to be the "win" and the other to be the "lose". What if finetune with RankDPO but using existing preference datasets? For example, set the "win" sample to be rank 1 and the "lose" sample to be rank 2. In my opinion, these insight is super important to this paper and would be beneficial to the community.

**Questions:**

No

---

> ### Author Response · Authors · 2024-11-20
>
> We thank the reviewer for taking the time to provide detailed feedback on our work. We also thank them for appreciating our ranking-based preference datasets, and our experiments. Below, we address the questions from the reviewer.
>
> **Results of DPO on Syn-Pic:** While Tab. 1,2,3 compare our preference-tuned SDXL and SD3-Medium models with other popular public models, in Tab. 4, all the methods are trained on the same data (i.e., Syn-Pic except the entry DPO (Pick-a-Picv2) which uses the Pick-a-Picv2 dataset). We see consistent gains from a) higher-quality images compared to Pick-a-Picv2 dataset, b) stronger and more consistent labeling through our reward model ensemble, and c) from the RankDPO objectives (last two lines).
>
> | **Model Name**         	| **DSG Score** | **VQA Score** | **Q-Align Score** |
> |-----------------------------|---------------|---------------|-------------------|
> | SDXL                   	| 74.65     	| 84.33     	| 0.72          	|
> | DPO (Random Labelling) 	| 75.66     	| 84.42     	| 0.74          	|
> | DPO (HPSv2)            	| 78.04     	| 86.22     	| 0.83          	|
> | DPO (Pick-a-Picv2)     	| 76.74     	| 85.67     	| 0.74          	|
> | DPO (5 Rewards)        	| 78.84     	| 86.27     	| 0.81          	|
> | **RankDPO**            	| 79.26 	| 87.52 	| 0.81      	|
>
>
> **MaPO with Ranking based Preferences:** An important aspect of our RankDPO formulation is that it is directly applicable to any pairwise preference-optimization loss (e.g., MaPO). Therefore, we can derive a **RankMaPO** formulation by applying the DCG and gain weights to the MaPO (or any other pairwise preference) objective:
>
> $$
> \Delta_{i,j} = |G_i - G_j| \cdot \left| \frac{1}{D(\tau(i))} - \frac{1}{D(\tau(j))} \right|.
> $$
>
> $$
> \mathcal{L}{\text{RankGeneralized}}(\boldsymbol{\theta}) = -\mathbb{E}{(\boldsymbol{c}, \boldsymbol{x}^1, \boldsymbol{x}^2, \dots, \boldsymbol{x}^k) \sim \mathcal{D}, t \sim [0,T]}
> \left[
> \sum_{i > j} \Delta_{i,j} \cdot \mathcal{L}_{\text{pairwise}}\left(s(\boldsymbol{x}^i, \boldsymbol{c}, t, \boldsymbol{\theta}), s(\boldsymbol{x}^j, \boldsymbol{c}, t, \boldsymbol{\theta})\right)
> \right].
> $$
>
> Such a formulation allows us to extend other preference objectives, such as MaPO, to include ranking-based cues to improve the optimization. In the case of SPO, it only takes prompts as input (similar to the other reward fine-tuning approaches such as AlignProp and TextCraftor). Therefore, it cannot utilize the images or labels from our Syn-Pic dataset. We elaborate on the RankMaPO formulation in our updated paper (Section A.3) and hope that our ranked preference dataset can improve various diffusion preference optimization methods.
>
>
> **RankDPO on binary preferences:** RankDPO could be applied to the binary preference setting where we have the winning and losing pair with ranks 1 and 2, respectively. On simplifying the RankDPO objective, we see that the DCG weights would be constant for the binary case. We would only be left with the gain weighting term, which we also show provides some improvements over the standard DPO formulation in Tab. 5 (second last row). However, in the Pick-a-Pic dataset, where we don't have reward model scores to construct the gain weights, our RankDPO formulation would entirely simplify into the standard DPO objective. We have a discussion about this in the updated paper (Sec A.1).

---

> > ### Author Response · Authors · 2024-11-25
> >
> > Dear Reviewer Rfve,
> >
> > We would like to thank you again for your valuable feedback on our paper. We have added clarifications to disentangle gains from Syn-Pic and RankDPO datasets. We have shown that both our components result in significant gains over existing datasets (such as Pick-a-Picv2) and methods (such as DPO). We have also included extensions/reductions of our work to existing direct preference frameworks.
> >
> > We have also included other experiments to show the benefits of our approach (see reply in main threads https://openreview.net/forum?id=Y6KUBkUimC&noteId=t0TJetYvG9 ).
> >
> > As the period for the Author-Reviewer discussion is closing very soon, we would like to use this opportunity to kindly ask if our responses sufficiently clarify your concerns. We sincerely appreciate your time and consideration.
> >
> > Best Regards, Authors

---

> > > ### Comment · Reviewer_Rfve · 2024-11-27
> > >
> > > Thank you for the response. Most of my concerns have been addressed. After reading the author response and comments from other reviewers, I lean towards keep my current positive rating for the following reason:
> > >
> > > * The rank-based dpo for diffusion model is interesting. Considering the result gain achieved by the proposed method, and the proposed reward dataset. I think this paper deserves a positive score in my opinion.
> > > * I am not convinced to give this paper a higher score. I agree with reviewer Pz5f's opinion that this paper is somewhat similar to a combination of diffusion-dpo and list-based dpo, making limited technical contribution.

---

### Author Response · Authors · 2024-11-20

We thank the reviewers for providing valuable feedback on our paper. We are pleased that they find our paper to have solid contributions (Reviewer GRrX), clear writing (Reviewers Rfve, GRrX),  and providing an effective solution (Reviewers 9PJP, Pz5f).  While we have addressed individual concerns in our replies, we outline the key additions to the paper below:

**a) Additional Baselines (Pz5f):** We have added PixArt-Sigma and Stable Cascade to Tab. 1,2,3 completing all the models that we use for generating Syn-Pic. We continue to observe that RankDPO applied to SDXL and SD3-Medium obtains strong results compared to other methods.


**b) FID-based evaluation (Pz5f):** We already evaluate our methods against baselines on several popular quantitative benchmarks (Tab. 1,2,3) and through user-studies (Fig. 3). In addition, in Tab. 10, we have included an evaluation on the MJHQ-30K set using FID against Midjourney images as an additional evaluation metric to measure image quality. We observe consistent improvements in FID scores as compared to existing results with SDXL. Although, it is worth noting that FID scores are sensitive to the choice of reference images and the pre-processing of features.
| **Category**   	| **Animals** | **Art**   | **Fashion** | **Food**  | **Indoor** | **Landscape** | **Logo**   | **People** | **Plants**  | **Vehicles** | **Overall** |
|---------------------|-------------|-----------|-------------|-----------|------------|---------------|------------|------------|-------------|--------------|-------------|
| **SDXL-Refiner**	| 28.93   	| 31.05 	| 28.90   	| 30.09 	| 28.83  	| 30.78     	| 36.67  	| 35.56  	| **28.42**   | 24.45    	| 9.55    	|
| **RankDPO SDXL**	| **24.37**   | **27.22** | **22.91**   | **25.02** | **26.01**  | **28.61** 	| **29.74**  | **27.54**  | 30.27   	| **21.83**	| **7.99**	|



**c) Clarify RLHF formulation (9PJP):** We now have a more detailed introduction of RLHF (Sec. 3.1) explaining the objective as well as the role of the reference model.


**d) Reward Model Strengths and Limitations (GRrX, 9PJP):** We now have a more detailed explanation for the reward models (Sec. 5), including discussing their high correlation to overall human judgment on various benchmarks, making them an ideal choice for labeling our preferences.


**e) Disentangling gains from Syn-Pic and RankDPO (Rfve, Pz5f):** Our ablations in Tab. 4 and 6 aim to break-down the gains from our proposed preference dataset and learning objective.
In Tab. 4, DPO (Pick-a-Picv2) vs DPO (5 rewards) compares the performance of DPO when trained on Pick-a-Picv2 and Syn-Pic datasets. It shows that training on the Syn-Pic dataset yields considerable improvements over training on Pick-a-Picv2.
Further, in Tab. 4, DPO (5 rewards) vs RankDPO compares the performance of RankDPO vs DPO when trained on the same Syn-Pic dataset, where we observe improvements over DPO. Additionally, in Tab. 6, we train DPO and RankDPO on the Syn-Pic dataset and evaluate their performance on GenEval benchmark. It shows that RankDPO consistently outperforms DPO.

| **Model Name**         	| **DSG Score** | **VQA Score** | **Q-Align Score** |
|-----------------------------|---------------|---------------|-------------------|
| SDXL                   	| 74.65     	| 84.33     	| 0.72          	|
| DPO (Pick-a-Picv2)     	| 76.74     	| 85.67     	| 0.74          	|
| DPO (5 Rewards)        	| 78.84     	| 86.27     	| 0.81          	|
| **RankDPO (Only SDXL)** 	| 78.40 	| 86.76 	| 0.74      	|
| **RankDPO**            	| 79.26 	| 87.52 	| 0.81      	|


| **Model**    	| **Mean ↑** | **Single ↑** | **Two ↑** | **Counting ↑** | **Colors ↑** | **Position ↑** | **Color Attribution ↑** |
|-------------------|------------|---------------|------------|-----------------|---------------|-----------------|--------------------------|
| SDXL         	| 0.55   	| 0.98      	| 0.74   	| 0.39        	| 0.85      	| 0.15        	| 0.23                 	|
| DPO	| 0.59   	| 0.99      	| 0.84   	| 0.49        	| 0.87      	| 0.13        	| 0.24                 	|
| **RankDPO**   	| **0.61**   | 1.00  	| 0.86   | 0.46    	| 0.90  	| 0.14    	| 0.29             	|
|-------------------|------------|---------------|------------|-----------------|---------------|-----------------|--------------------------|
| SD3-Medium   	| 0.70   	| 1.00      	| 0.87   	| 0.63        	| 0.84      	| 0.28        	| 0.58                 	|
| DPO	| 0.72   	| 1.00      	| 0.90   	| 0.63        	| 0.87      	| 0.32        	| 0.58                 	|
| **RankDPO**   	| **0.74**   | 1.00  	| 0.91   | 0.72    	| 0.87  	| 0.31    	| 0.66             	|

---

### Meta-Review · Area_Chair_Hv66 · 2024-12-21

**Metareview:**

This paper has two major contributions: Syn-Pic, a synthetic preference dataset collection method, where preferences are generated by using multiple existing reward models; RankDPO, a extended DPO method with list-wise preference.

Major paper strengths: 1) using existing reward models to create synthetic list-wise preference data and  applying RankDPO to this listwise preference data is a reasonable solution 2) Experiments on multiple diffusion models like SDXL and SD3-Medium on multiple benchmarks shows the effectiveness of the proposed approach.

Major weaknesses: 1) As pointed out by reviewer Pz5f and Rfve, the technical novelty of this paper is not high. RankDPO is essentially similar as LiPO: Listwise Preference Optimization through Learning-to-Rank (https://arxiv.org/html/2402.01878v1) . And the idea of using existing reward models to generate synthetic preference data is straightforward. 2) The motivation of using existing reward models in Syn-Pic is to eliminate human labeled data, however, human labeled data are needed when creating the reward models. So it does not really eliminate human labeled data, but instead implicitly leverage existing human labeled data in previous reward models. Moreover, existing reward models may not generalize well to future generative models (when model quality and data distribution are different). So this method will not eliminate human labeled data for future generative models either.

Given the two major weakness and the overall average score of this paper is 6, not very high, I will recommend "Reject".

**Additional Comments On Reviewer Discussion:**

Both reviewer Pz5f and Rfve are not convinced by the authors' rebuttal on novelty and feel the proposed method is a combination of diffusion-dpo and list-based dpo (LiPO). And I shared the same feeling.

Reviewer GRrX asked about evaluation benchmarks and reward models, which is addressed well by the authors.

---

### Decision · Program_Chairs · 2025-01-22

Reject